

# Assessment of tropospheric CALIPSO Version 4.2 aerosol types over the ocean using independent CALIPSO-SODA lidar ratios

Zhujun Li[1,2], David Painemal[1,2], Gregory Schuster[2], Marian Clayton[1,2], Richard Ferrare[2], Mark Vaughan[2], Damien Josset[3], Jayanta Kar[1,2], Charles Trepte[1]

[1]Science Systems and Applications Systems, Inc.
[2]NASA Langley Research Center
[3]US Naval Research Laboratory Stennis Space Center, MS 39529, USA

*Correspondence to*: david.painemal@nasa.gov

**Abstract.** We assess the CALIPSO Version 4.2 (V4) aerosol typing and assigned lidar ratios over ocean using aerosol optical depth (AOD) retrievals from the Synergized Optical Depth of Aerosols (SODA) algorithm and retrieved columnar lidar ratio estimated by combining SODA AOD and CALIPSO attenuated backscatter (CALIPSO-SODA). Six aerosol types – clean marine, dusty marine, dust, polluted continental/smoke, polluted dust, and elevated smoke – are characterized using CALIPSO-SODA over ocean and the results are compared against the prescribed V4 lidar ratios, when only one aerosol type is present in the atmospheric column. For samples detected at 5-km or 20-km spatial resolutions and having AOD > 0.05, the CALIPSO-SODA lidar ratios are significantly different between different aerosol types, and are consistent with the type-specific values assigned in V4 to within 10 sr (except for polluted continental/smoke). This implies that the CALIPSO classification scheme generally categorizes aerosols correctly. We find remarkable daytime/nighttime regional agreement for clean marine aerosol over the open ocean (CALIPSO-SODA = 20-25 sr, V4=23 sr), elevated smoke over the southeast Atlantic (CALIPSO-SODA = 65-75 sr, V4=70 sr), and dust over the subtropical Atlantic adjacent to the African continent (CALIPSO-SODA = 40-50 sr, V4=44 sr). In contrast, daytime polluted continental/smoke lidar ratio is more than 20 sr smaller than the constant V4 vaue for that type, attributed in part to the challenge of classifying tenuous aerosol with low signal-to-noise ratio. Dust over most of the Atlantic Ocean features CALIPSO-SODA lidar ratios less than 40 sr, possibly suggesting the presence of dust mixed with marine aerosols or lidar ratio values that depend on source and evolution of the aerosol plume. The new dusty marine type introduced in V4 features similar magnitudes and spatial distribution as its clean marine counterpart with lidar ratio differences of less than 3 sr, and nearly identical values over the open ocean, implying that some modification of the classification scheme for the marine subtypes is warranted.

## 1. Introduction

For more than 15 years, the Cloud Aerosol Lidar with Orthogonal Polarization (CALIOP), onboard the Cloud-Aerosol Lidar and Infrared Pathfinder Satellite Observation (CALIPSO) platform, has revolutionized our understanding of the role of aerosols in the climate system, revealing little known aspects of long-range aerosol transport, as well as the aerosol structure in the boundary layer and the free troposphere (e.g., Adams et al., 2012; Winker et al., 2013; Yu et al., 2015; Kacenelenbogen et al., 2019; Jumelet et al., 2020). CALIOP observations have also enabled one of the most comprehensive aerosol validation efforts of chemical transport models, with the focus on



aerosol vertical structure (e.g. Koffi et al., 2012; 2016). With a lifespan that has far exceeded the design specifications, CALIPSO has set the foundations for future spaceborne lidar missions and will remain as the longest data record of global vertically resolved aerosol properties for many years to come. Thus, it is essential to continue improving CALIPSO retrievals to provide increasingly accurate benchmarks for assessing the role of various aerosol types at different spatiotemporal scales, and for climate model evaluation.

The primary challenge of deriving aerosol extinction coefficient and aerosol optical depth using CALIOP (or any other elastic-backscatter lidar) is to separate the particulate backscatter and extinction coefficients with only one direct measurement: the attenuated backscatter coefficients. To resolve this ambiguity, CALIPSO adopts the standard procedure of reducing the inversion problem to one unknown by relating extinction and backscatter coefficients in the lidar equation via an assumed extinction-to-backscatter ratio (or lidar ratio, Young and Vaughan, 2009). Since aerosol

types can be characterized by specific lidar ratios (e.g., Müller et al., 2007), CALIPSO first classifies aerosols into several categories and assigns predetermined lidar ratios to each aerosol class. The aerosol typing algorithm and lidar ratio selection was originally based on a cluster analysis of a multiyear Aerosol Robotic Network (AERONET) dataset (Omar et al., 2009) that classifies tropospheric aerosols into six categories. In the latest version of CALIOP algorithm (CALIOP V4), aerosols are classified into seven subtypes in the troposphere and four subtypes in the stratosphere (Kim

et al., 2018). The newly added tropospheric subtype is dusty marine, which represents a mixture of dust and marine aerosols and was developed to accommodate Saharan dust entrained into the boundary layer over the Atlantic Ocean (e.g. Groß et al., 2016). Finally, CALIOP V4 lidar ratios for dust, elevated smoke, clean marine, and clean continental have been revised from earlier versions, based either on advanced retrieval techniques developed since the launch of the mission (for dust and smoke; see Liu et al., 2015) or by compositing multiple measurements acquired by high

spectral resolution lidar (HSRL) and Raman lidar. Because the clean continental tropospheric aerosol type is only identified over land, we necessarily omit it from the ocean-only analyses presented in the remainder of this paper.

       Briefly, the six tropospheric aerosol types over the ocean are determined based on integrated attenuated backscatter ($\gamma$), estimated particulate depolarization ratio ($\delta$), the aerosol layer top and base altitude ($Z_{top}$, $Z_{base}$), and the surface type. The specific typing thresholds and associated CALIPSO V4 lidar ratios for aerosols over the ocean

are summarized in Table 1. Boundary layer and lower tropospheric aerosols over the ocean are classified into non-depolarizing clean marine, weakly scattering and mildly depolarizing polluted continental/smoke, and dusty marine (a moderately depolarizing mixture). Dust is characterized by its high particulate depolarization ratio. Moderately depolarizing polluted dust over oceans and non-depolarizing elevated smoke over all surface types are, by definition, only identified for aerosols having layer top altitudes at or above 2.5 km. Polluted dust also includes moderately

depolarizing aerosols and weakly scattering non-depolarizing aerosols over desert land. The CALIOP observables do not contain sufficient information to reliably distinguish between pollution plumes and low-lying smoke layers. Thus, the main difference between polluted continental/smoke and elevated smoke is that they are defined, respectively as having layer tops below and above 2.5 km.

       The aforementioned updates implemented in CALIPSO V4 yield better agreement with AERONET and MODIS

aerosol optical depth (AOD) than CALIPSO Version 3 (Kim et al., 2018). However, biases are likely to persist for



some specific aerosol types and regions. For instance, Painemal et al. (2019) found substantial overestimations of CALIOP V4 aerosol extinction coefficients relative to airborne HSRL measurements over the Caribbean. Understanding biases in AOD and lidar ratios is challenging, as two main factors can cause discrepancies: a) the ability of CALIPSO algorithm to detect and classify aerosol layers, and b) uncertainty in the prescribed lidar ratio used for the CALIPSO inversion. While layer under-detection causes a well-known systematic underestimation in AOD (Toth et al., 2018; Kim et al., 2017), uncertainties attributed to the assigned lidar ratio and the algorithm's ability to correctly identify the aerosol type can be manifested in an underestimation or overestimation of AOD. Given the large impact of the lidar ratio choice on the retrievals (Winker et al., 2009), it would be desirable to directly assess the lidar ratios used by CALIPSO with independent measurements. Lidar ratios derived from HSRL and Raman lidar are the best available datasets for such studies (e.g. Rogers et al., 2014; Burton et al. 2013; Müller et al., 2007; Groß et al., 2013; Wang et al., 2021), yet their spatiotemporal coverage is extremely limited and the sampling is dissimilar to CALIOP.

In this study, we compare CALIPSO version 4.2 aerosol products and lidar ratios to a CALIPSO-based research product: the CALIOP Synergized Optical Depth of Aerosols (CALIPSO-SODA). We derive the CALIPSO-SODA lidar ratios by applying a Fernald-Klett inversion (Fernald, 1972; Fernald et al., 1984; Klett, 1985) to the CALIOP attenuated backscatter coefficients and the SODA AODs. Our goal is to determine how well the prescribed CALIPSO V4 lidar ratios compare to the retrieved CALIPSO-SODA lidar ratios for each CALIPSO aerosol type over the ocean.

## 2. Data and method

### 2.1 CALIPSO V4 and SODA data

We use daytime and nighttime CALIPSO Level 2 Aerosol Profile (APro) version 4.2, with an effective horizontal resolution of 5 km. However, we note that CALIPSO aerosol classifications are determined using attenuated backscatter averaged at 5, 20, or 80-km resolution, depending upon the tenuousness of the aerosol feature. The role of the spatial averaging on the aerosol classification will be further analyzed in Section 3. We only retain cloud-free observations with no stratospheric features, as our focus is on tropospheric lidar ratios. In addition to aerosol type and spatial averaging information, APro also provides aerosol-cloud classification quality flags that are applied to minimize retrieval uncertainties (Section 2.3). Lastly, cloud mask is taken from Level 2 Vertical Feature Mask (VFM), with a horizontal resolution of 333 m and re-gridded to 5 km resolution. Cloud-free scenes are determined from the VFM cloud mask with 333 m horizontal resolution below 8.2 km and 1 km above.

### 2.2. CALIPSO-SODA lidar ratio

Here, we briefly summarize the CALIPSO-SODA lidar ratio retrieval algorithm described in Painemal et al., (2019). CALIPSO-SODA lidar ratios are estimated using the Fernald-Klett inversion method by constraining the lidar equation with SODA AOD (Josset et al., 2015), and relating aerosol backscatter and extinction coefficients via the lidar ratio. SODA AOD is a research product derived from the CALIOP and CloudSat Cloud Profiling Radar surface returns over water and it has shown good agreement with airborne lidar observations and MODIS AOD (Painemal et al., 2019; Josset et al., 2015). While other satellite AOD products can be combined with CALIOP to derive lidar ratio,



SODA AOD is by definition collocated with CALIPSO and retrievals are possible during both daytime and nighttime overpasses. In short, the inversion algorithm consists of assuming an initial lidar ratio and solving the lidar equation for aerosol extinction coefficient in the profile. Next, the lidar ratio is iteratively adjusted until the retrieved AOD

(estimated by vertically integrating the retrieved extinction profile) matches the SODA AOD. Note that the same lidar ratio is applied to all aerosol layers in the region of interest, so this method retrieves a single effective lidar ratio for a given optical depth constraint. We can include more than one lidar ratio in a column, however, by assuming a fixed lidar ratio for a portion of the atmosphere. This is described next.

Painemal et al., (2019) uses two different CALIPSO-SODA lidar ratio techniques that are based upon 2 different

assumptions: 1) the one-layer technique (1L), which assumes one lidar ratio in the aerosol column; and 2) the two-layer technique (2L), which prescribes the lidar ratio in the marine boundary layer at 25 sr and iteratively calculates the lidar ratio for the column above the boundary layer. Thus, the 2L technique is applied irrespective of the occurrence of the Clean Marine type in V4. The boundary layer height is estimated from meteorological analysis using the bulk Richardson number (e.g. McGrath-Spangler and Molod, 2014). As shown in Figure 1, the marine atmospheric

boundary layer (MABL) varies between 300 m and 700 m over the open ocean, with minima over the equator, and heights ranging between 700-1000 m near the coasts during daytime. The 2L assumption is expected to best represent cases of aerosol layers predominant in the free troposphere, such as smoke and dust. More specifically, the average dust base height over the ocean of 961 m (daytime) and 438 m (nighttime) are close to the MBL height in Figure 1, further justifying the use of 2L lidar ratio for characterizing dust over the open ocean. Daytime and nighttime retrievals

are limited to the 2006-2011 period because CloudSat (required for deriving SODA AOD) switched to daytime-only operation after October 2011 due to a battery malfunction. Since previous comparisons with airborne HSRL, during both day and nighttime, have shown that the SODA AOD uncertainty is 0.035 (Painemal et al., 2019), we only perform the analysis on profiles with SODA AOD > 0.05, to avoid retrieving lidar ratios when the SODA AOD is lower than this uncertainty.

We compute lidar ratios by applying the Fernald-Klett algorithm to truncated CALIOP attenuated backscatter profiles. We truncate the profiles at high altitudes to minimize uncertainties associated with weak signal-to-noise ratios, which typically occur at high altitudes where the aerosol layers are too tenuous to be detected by CALIPSO. The truncation altitude, also implemented in Painemal et al. (2019), is derived by using CALIPSO VFM to find the highest altitude in the profile where CALIPSO detects an aerosol layer ($VFM_{MAX}$). Being cognizant that the low signal-to-noise

ratio (SNR) of the CALIPSO measurements often causes diffuse aerosol layers to go undetected by the CALIPSO retrieval algorithms (Kim et al., 2017; Toth et al., 2018), we define the maximum altitude in the CALIPSO-SODA algorithm as 2 km above $VFM_{MAX}$. It follows that for columns with no detected aerosol layers, the maximum CALIPSO-SODA altitude is 2 km above the sea level. We chose $VFM_{MAX}$+2km because this yields the best agreement between CALIPSO-SODA and HSRL in a previous comparison (Painemal et al., 2019), with a root-mean-squared

error (RMSE) of 7.4 sr and a negative bias of -2.5 sr and -4.7 sr for 1L and 2L CALIPSO-SODA, respectively. We have also tested the use of $VFM_{MAX}$+1km and $VFM_{MAX}$+3km and found that $S_a$ slightly decreases as the maximum altitude increases. More specifically, the daytime lidar ratios decrease by an average of -1.3 sr when the truncation



altitude is increased from VFMmax+2km to VFMmax + 3km, and the same lidar ratios increase by an average of +1.6 sr when the truncation altitude decreases from $VFM_{MAX}$+2km to $VFM_{MAX}$+3km. Nighttime differences in lidar ratio differences are smaller at 0.9 sr ($VFM_{MAX}$+1km) and -0.75 sr ($VFM_{MAX}$+3km) relative to $S_a$ for $VFM_{MAX}$+2km.


A more comprehensive analysis of the effect of truncating the attenuated backscatter is presented next. In agreement with the analysis above, Kim et al. (2016) noted that retrieved lidar ratios derived using CALIPSO data constrained with MODIS AOD decrease with the initial iteration altitude. Following Kim et al., (2016), we look at the iteration height in CALIPSO-SODA by comparing our $VFM_{MAX}$+2 km assumption against retrievals that also make use of SODA AOD but estimated using as the initial height: a) the tropopause height according to GEOS-5, and b) 36-km height, right below the 36-39 km calibration layer used in V4. Figure 2 shows that the lidar ratios resulting from iterating to higher altitudes are smaller than the ones generated using the VFM-based assumption; the discrepancy can be as much as ~70 % relative to the lidar ratios used in this study, and thus, substantially underestimating the HSRL retrievals in Painemal et al., (2019). These differences are substantially smaller for nighttime and with higher linear correlation coefficient ($r$) due to stronger SNR.



It is important to indicate that we have ascribed the total AOD (from SODA) to the truncated profiles, under the assumption that layer AOD for altitudes above $VFM_{MAX}$+2-km can be neglected. This assumption can, to some degree, be evaluated by considering a climatology of stratospheric AOD. For this, we use the aerosol extinction measurements from Stratospheric Aerosol and Gas Experiment III (SAGE-III) onboard the International Space Station (ISS). SAGE-III is the latest in the SAGE series of instruments which have been providing the most accurate stratospheric aerosol measurements using the occultation technique, since 1984. Aerosol extinction profiles and stratospheric AOD from SAGE-III are available since June 2017 in nine channels from 384 nm to 1544 nm including one at 521 nm (Cisewski et al., 2014). We have used an Angstrom exponent of 1.6 to convert 521 nm stratospheric AOD to 532 nm (version 5.2) and have applied a fractional uncertainty filter of 50% to AOD. Figure 3 shows the spatial distribution of the stratospheric AOD at 532 nm from SAGE-III for the time period June 2017 through April 2021. Several volcanic events like Ambae, Raikoke and Ulawun as well as the strong pyroCb events of August-September 2017 in Canada and January-February 2020 in Australia significantly perturbed the stratosphere during this period. These perturbations persist for many months to a year in the stratosphere (e.g. Kloss et al., 2021) and removing them would significantly reduce the sampling. We have therefore retained all of the data in this climatological map and the AOD values may represent stratospheric loading above the "background". This gives a global mean stratospheric AOD of ~0.01 and 0.007 over the subtropics. This stratospheric AOD is less than 10% of the mean SODA AOD. To put this result in context, Painemal et al., (2019) noted that a 20 % overestimation in AOD would impact CALIPSO-SODA lidar ratio producing uncertainties of +6 sr. Even though systematic biases are introduced when using a truncated profile, these errors are significantly smaller than the uncertainties in the lidar ratios prescribed by the CALIPSO algorithm (Table 1).





CALIPSO-SODA extinction coefficients (for both 1L and 2L assumptions) were evaluated against HSRL observations over the western Atlantic in Painemal et al., (2019), yielding high linear correlations with airborne HSRL measurements (> 0.7) over the western Atlantic and negative mean biases around -2.5 to - 4.7 sr (1L and 2L,



respectively). In contrast, the standard CALIPSO V4 product overestimated the boundary layer extinction coefficient by up to 100% at night and 140% during the day. The good agreement between CALIPSO-SODA and HSRL measurements is also manifested in the lidar ratio, with CALIPSO-SODA biases as low as -2.5 sr.

Lastly, CALIPSO-SODA retrievals used here and described in Painemal et al. (2019) are derived at 1-km. Thus, the matching between CALIPSO-SODA lidar ratio and V4 parameters is done by averaging the 1-km CALIPSO-185    SODA to the 5-km native resolution of V4. As CALIPSO aerosol typing is performed using different spatial averaging, it is of interest to determine whether CALIPSO-SODA lidar ratio is sensitive to the spatial resolution of the lidar attenuated backscattering coefficient (β') ingested in the Fernald algorithm. To address this, we averaged β' to achieve 5, 20, and 80 km spatial resolution and compared the retrieved lidar ratios against its 1-km counterpart averaged to the corresponding spatial resolution. The sensitivity analysis based on 5 days of daytime and nighttime observations of 190    July 2010 reveals mean differences between 1-km $S_a$ and its coarser-resolution counterparts are less than -1.42 sr for daytime and 0.81 sr for nighttime, with linear correlation coefficients greater than 0.83 (Figure A1). We conclude that for the purpose of this analysis, the effect of different β' spatial averaging is minimal and, therefore, the CALIPSO-SODA lidar ratio has little sensitivity to the spatial resolution used by V4 to perform the spatial averaging.

## 2.3. CALIPSO-SODA and V4 screening and methodology

The main obstacle to comparing CALIPSO-SODA lidar ratios to CALIPSO V4 retrievals is that CALIPSO-SODA is a column quantity with a single lidar ratio, whereas V4 profiles can accommodate multiple aerosol types and multiple lidar ratios. To avoid the ambiguity of relating a single lidar ratio to a CALIPSO profile that can feature more than one aerosol type, we analyze CALIPSO retrievals when only one tropospheric aerosol type is identified in the 200    column (excluding clear air), irrespective of whether V4 is compared against 1L or 2L CALIPSO-SODA retrievals. The CALIPSO aerosol types over the ocean are clean marine, dust, polluted continental/smoke, polluted dust, elevated smoke, and dusty marine. In addition to the data screening described in Section 2.1 and 2.2 (AOD>0.05), we restrict the analysis to V4 profiles with a Cloud-Aerosol Discrimination (CAD) score > |-50| (moderate to high confidence of aerosol classification). A further constraint adopted in this study is that the entire vertical column must be free of clouds 205    over the 5-km horizontal resolution. This screening is adopted because CALIPSO-SODA retrievals are limited to cloud-free conditions; screening minimizes cloud contamination in the retrievals and reduces the swelling effects of aerosols near the cloud edges (Varnai and Marshak, 2018). Finally, given that CALIPSO aerosol typing depends on the aerosol layer height (Table 1), we characterize clean marine, dusty marine, and polluted continental smoke using CALIPSO-SODA lidar ratios based on the 1L assumption; dust, polluted dust, and elevated smoke aerosols are 210    described by means of the CALIPSO-SODA 2L assumption, to isolate the lidar ratios from elevated layers from those in the boundary layer (likely dominated by marine aerosols).

Global maps of daytime number of samples analyzed in this study are summarized in Figure 4. The largest number of samples correspond to marine aerosol, which shows a relatively homogeneous spatial distribution. The new dusty marine category features the second largest number of samples, with the highest density primarily confined to 215    the subtropics. Dust, unsurprisingly, prevails over the tropical Atlantic Ocean, in connection with the westward





transport from the Saharan Desert, and over the Arabian and Mediterranean Sea (e.g. Kaufman et al., 2005). Polluted continental/smoke features regional peaks in the southeast Pacific, and the eastern Atlantic. The relatively large number of samples (also observed at nighttime, Figure A2) of polluted continental/smoke over the open ocean is somewhat unexpected, especially over the southeast Pacific, where the anticyclonic circulation tends to confine the transport of
continental aerosols to the coastal domain (Yang et al., 2011). It is interesting that the spatial distribution of polluted continental/smoke and dusty marine are qualitatively similar. Considering that both aerosol types occur in the boundary layer and with a unique depolarization ratio threshold (0.075) that separates them (Table 1), their spatial distribution suggest that a selection of a smaller threshold for polluted continental will result in both a reduced number of samples classified as polluted aerosols and an increase of dusty marine over the open ocean. Elevated smoke reaches a maximum
over the southeast Atlantic Ocean, in connection with the biomass burning season of southern and equatorial Africa (Roberts et al., 2009, Redemann et al., 2021). Lastly, polluted dust resembles the spatial distribution of elevated smoke, which reflects the influence of biomass burning emissions and that both aerosol types are defined for aerosol plume elevations above 2.5 km a.m.s.l. (Table 1).

**2.4. SODA and CALIPSO V4 AOD**

Before presenting the lidar ratio analysis, it is pertinent to compare SODA and CALIPSO V4 AOD for different aerosol types over the ocean. Studies that have compared SODA AOD against aircraft data, MODIS, and the POLarization and Directionality of the Earth's Reflectance (POLDER) on board Parasol satellite show good agreement (Josset et al., 2015). Similarly, Painemal et al., (2019) found SODA AOD regional biases smaller than those observed
for CALIPSO AOD. In the context of this study, SODA AOD is expected to be more accurate than V4, and therefore, differences between both datasets primarily reflect uncertainties in V4. The SODA-V4 comparison also offers a glimpse of the expected differences in lidar ratio between these two products. For instance, SODA AOD greater than its V4 counterpart would yield greater SODA-based lidar ratio than that prescribed by V4.

Here, we define the bias as the mean difference between V4 AOD ($AOD_{V4}$) and SODA AOD ($AOD_{SODA}$) or:

$$Bias = \frac{1}{N}\sum_{i=1}^{N} AOD_{V4} - AOD_{SODA} \quad (1)$$

Root mean square error (RMSE) is calculated as:

$$RMSE = \sqrt{\frac{1}{N}\sum_{i=1}^{N}(AOD_{V4} - AOD_{SODA})^2} \quad (2)$$

Both bias and RMSE are also expressed in terms of % relative to the mean $AOD_{SODA}$ ($\overline{AOD_{SODA}}$) as: $\frac{Bias}{\overline{AOD_{SODA}}} X 100\%$ and $\frac{RMSE}{\overline{AOD_{SODA}}} X 100\%$.

The SODA and CALIPSO V4 AOD comparison for each aerosol type is depicted in Figures 5 and 6, with statistics summarized in Table 2. In general, AOD for polluted continental/smoke is the smallest among the six aerosol types, while dust and elevated smoke have the largest AOD (open black circles). The linear correlation coefficient ($r$)
between CALIOP V4 and SODA AOD is the highest for dust, with $r = 0.59$ (daytime) and 0.63 (nighttime). For other



aerosol types the correlations are modest, with negligible values for polluted dust primarily explained by its narrow AOD dynamic range. V4-SODA linear correlation coefficient is the highest for dust, suggesting that this aerosol type is best classified by the V4 algorithm, which is aided by the high depolarization ratio signature of dust. In terms of the V4 biases, V4 is systematically greater than SODA for dust, with biases that increase with AOD, particularly during

nighttime, and biases of 45 % relative to the mean SODA AOD. Similarly, elevated smoke V4 AOD overestimates SODA by 128% during nighttime. By contrast, V4 AOD is smaller than its SODA counterpart for daytime polluted dust (59%), possibly contributed by the layer below 2.5 km, which is not accounted for in V4, as only elevated aerosols are classified as polluted dust in V4 (Table 1). As discussed in the Introduction, the primary causes for the observed departures of V4 AOD are misclassification of aerosol types and tenuous aerosol layers that are not detected by the

CALIPSO algorithm, with the latter contributing to an underestimation of V4 AOD. Lastly, AOD differences stemmed from lidar ratio differences will be further explored in the following.

## 3. Lidar ratio statistics

We use the standard CALIPSO V4 aerosol typing to group our retrieved lidar ratios into six aerosol types (Kim

et al., 2018), with the methodology and CALIPSO-SODA lidar ratio algorithm described in section 2.2. The results are shown in Table 3 as well as Figures 7 and 8. We first analyze the daytime lidar ratio for samples classified by the V4 typing algorithm using all spatially averaged observations (at 5 km, 20 km, and 80 km, Figure 7a). Figure 7a reveals a degree of separation between some aerosol types, and some typing variability that is qualitatively consistent with expectations. For instance, clean marine (green) and dusty marine (brown) feature medians below 40 sr, whereas

elevated smoke presents the highest lidar ratios with a median around 50 sr.

We take a closer look at the effect of V4 spatial averaging by separating the samples into profiles retrieved exclusively with 80 km spatial averaging data and those with 20 km and/or 5 km resolution (Figures 7b and c, respectively). CALIPSO-SODA lidar ratios binned exclusively with 80-km aerosol typing depict similar medians with differences of less than 7 sr, suggesting a lack of skill of the typing algorithm for classifying aerosols at 80-km spatial

resolution. This could be in part caused by mixing of different aerosol plumes at such large horizontal scales, although a resolution of 80-km is generally smaller than the spatial variability of homogeneous aerosol layers over the ocean (e.g. Anderson et al., 2003). In contrast, we see a variety of median lidar ratios when the analysis is repeated for 5 km and/or 20 km (5+20 km) aerosol typing resolution; that is, small lidar ratios for clean marine and dusty marine, moderate magnitudes for dust, and large values for polluted dust, polluted continental, and elevated smoke. We also

find a variety of median lidar ratios for nighttime data, as shown in Figure 8. Here again, the CALIPSO-SODA retrieval for 80-km layers produces lidar ratios that are inconsistent with expectations (Fig 8b). For instance, CALIPSO-SODA lidar ratios for clean marine and dusty marine aerosols have medians around 60 sr, which is substantially higher than literature values for marine aerosols (e.g. Müller et al., 2007; Burton et al., 2012).

In light of these findings, the following analysis is conducted based on V4 aerosol classification determined

from 5-km and/or 20-km spatial averaging (unless otherwise indicated). A close look at both the daytime and nighttime lidar ratios in Table 3 reveals that the magnitudes are generally within 10 sr of the prescribed V4 values, except for



polluted continental/smoke and elevated smoke. Table 3 also indicates that nighttime lidar ratios for polluted continental/smoke (56 sr) and elevated smoke (47 sr) also depart from the value of 70 sr prescribed in V4 and are substantially different from their daytime counterparts. Discrepancies between CALIPSO-SODA and V4 can be further
understood in terms of their geographical distribution, as discussed in Section 4 below.

An additional lidar ratio analysis with a more stringent constraint is also performed by only selecting profiles classified by CALIPSO from 5-km horizontally averaged samples (Figure 9). The underlying assumption is that these 5-km samples offer the most suitable SNR conditions for aerosol classification. CALIPSO-SODA statistics listed in Figure 9 are only performed for marine aerosols (clean and dusty) and dust because they are the only types that yield
enough samples for a statistically robust analysis. The median dust lidar ratio remains around 34 sr during either daytime or nighttime, a range that is nearly 10 sr smaller than V4 assigned value. In terms of marine aerosol type, the daytime and nighttime medians (24-25 sr) are only a few steradians greater than the V4 counterpart for clean marine aerosols. Interestingly, the median differences between clean marine and dusty marine are modest (within 3 sr), and with substantial overlap in their inter-quartile ranges.
While the differences between lidar ratios for aerosols classified using different spatial averaging data is in part attributed to SNR changes, the role of the signal strength on the aerosol classification can be analyzed by investigating the dependence of lidar ratio for different values of SODA AOD. For this purpose, we stratify the data for three SODA AOD segments: 0.05-0.10, 0.10-0.15, and AOD > 0.15. Lidar ratio changes with AOD remain below 6 sr for clean marine, dusty marine, and dust (not shown) whereas polluted continental/smoke, elevated smoke, and polluted dust in
Figure 10 feature an increase with AOD of more than 10 sr during daytime, which shows a systematic increase with AOD to values of 58 sr for daytime and 72 sr for nighttime. This suggests that the derived lidar ratio for polluted continental type is in better agreement with the value used in V4 (70 sr).

## 4. Lidar ratio maps

Specific geographical occurrence for specific aerosol types have been documented in multiple studies, particularly for dust and smoke. It is, thus, pertinent to analyze the spatial distribution of CALIPSO-SODA lidar ratios to determine the extent over which the retrieved lidar ratios as a function of aerosol type are consistent with values in the literature for regions with dominant aerosol types. We present geographical maps of CALIPSO-SODA median
lidar ratios in Figures 11, 12, 14, and, 15 using 5+20-km resolution for the CALIPSO aerosol type. The maps are constructed using 10° x10° grid boxes and we only consider samples with AOD > 0.05, as in Section 3. Additionally, we only report grids containing medians estimated from at least 20 samples, which typically limits the median uncertainty to less than ±10 sr at the 95 % confidence interval according to the test described in Krzywinski and Altman (2014). We discuss the main results for each V4 aerosol type below.

### 4.1. Clean marine and dusty marine



Maps for the two marine aerosol types (Figure 11) are quite similar, with values ranging between 20 sr and 40 sr over the open ocean in the daytime and comparable magnitudes (if not smaller) for nighttime. While these relatively low lidar ratios are somewhat consistent with expectations, values near coastal regions are often significantly greater than over the open ocean. For instance, magnitudes between 40 and 50 sr are common off of the coast of Asia, and a peak of 55 sr is observed for clean marine aerosols over the Bay of Bengal. This is consistent with pollution being classified as marine aerosol in heavily polluted coastal areas. Median lidar ratios for marine aerosol samples with SODA AOD > 0.15 (Figures A3 and A4) for aerosol types feature values over the Southern Ocean of less than 25 sr. The occurrence of low lidar ratio south of 40°S is consistent with an increase in AOD associated with sea salt production driven by the strong surface zonal winds observed in satellite retrievals and shipborne observations (Wilson et al., 2010). In contrast, the coastal marine aerosol lidar ratios increase for AOD > 0.15 are likely the manifestation of pollution and continental aerosols advected to the adjacent ocean (Figures A3 and A4). Our results are qualitatively consistent with the SODA-based analysis of Dawson et al. (2015) for clean marine aerosol, even though their lidar ratios are smaller than those presented in Figure 11. Unlike the iterative method applied in our study, lidar ratios in Dawson et al. (2015) were derived using an analytical relationship based on the vertically integrated lidar equation (Platt, 1973). Comparisons between the iterative Fernald method and the vertically integrated equation (not shown) yield lidar ratios around 4 sr larger for the Fernald method, in agreement with the discrepancies between our study and Dawson et al. (2015).

In Figure 12 we repeat the lidar ratio maps for the marine aerosol types shown in Figure 11, but here we limit the data to only those layers detected at CALIOP's 5-km horizontal averaging resolution. The similarity between clean and dusty marine is remarkable over the open ocean and south of 40°S, where lidar ratios are around 20-25 sr and nearly identical to the value for clean marine used by V4 (23 sr). Clean and dusty marine aerosols also show comparable values over coastal regions for the 5-km aerosol layers, with lidar ratio peaks east of India and over the Arabian Sea (40-55 sr). Relatively large lidar ratios for marine aerosols near the coast points to aerosol misclassification or the presence of a mixture of marine and polluted aerosols. While dusty marine lidar ratios slightly exceed those for clean marine near the coast, consistent with expectations, the differences become negligible far offshore especially for 5-km data. The contrast between coastal and open-ocean samples is more clearly depicted in Figure 13, with coastal samples defined as being located within 5° from the coast, and offshore samples as those at least 10° away from the continents. Clean marine lidar ratio for coastal samples feature values near 28 sr but with upper quartile ≥ 35 sr for both daytime and nighttime, whereas its dusty marine counterpart yields median lidar ratios between 28-32 sr (daytime and nighttime) and upper quartile values of 40-42 sr. In contrast, the reduced interquartile variability for offshore samples relative to their coastal counterpart is evident, as well as the lower lidar ratio for offshore samples, which fluctuates between 23-26 sr for both marine aerosol types.

## 4.2. Dust and Polluted dust

Dust shows lidar ratio values ranging between 35 to 55 sr for daytime data and greater than 45 sr for the North American coast and the littoral zone of the northeastern Atlantic and the Mediterranean Sea (Figure 14, upper panels).



Nighttime dust is more variable than daytime dust, with lidar ratios up to 70 sr off of the west coast of Australia, and ranges between 35-45 sr over most of the Atlantic Ocean, where dust is the dominant species. It is, nevertheless,

puzzling that nighttime dust lidar ratio is smaller than its day counterpart, especially considering that nighttime conditions present more favorable conditions for retrieving optical properties from CALIPSO. Interestingly, V4 AOD for dust is much larger during nighttime (80% relative to the mean SODA AOD) than daytime (45%), implying that daytime to nighttime differences in CALIPSO-SODA lidar ratio are not necessarily attributed to uncertainties in the Fernald algorithm. Similarly, the 5-km dust maps (Figure 14, lower panels) also suggest lidar ratios between 35-45 sr

over the Atlantic Ocean, but with magnitudes closer to 55 sr near the African coast. These results are somewhat in agreement with the new value adopted by CALIPSO V4 (44 sr), which was revised after the analysis of Liu et al. (2015) for above-cloud aerosol layers over the north Atlantic Ocean. For polluted dust, meaningful numbers of samples are limited to daytime retrievals over the southeast Atlantic and western north Pacific (Figure 15). Polluted dust in Figure 15 shows a typical range of 45-65 sr, consistent with the assumed value in V4, with a clear regional peak over the

biomass-burning dominant southeast Atlantic.

Our CALIPSO-SODA lidar ratios for dust are within the middle to lower end of high spectral resolution (HSRL) and Raman lidar observations for regions adjacent to Saharan desert and Patagonia. From ship-borne measurements, Kanitz et al. (2013) document lidar ratios between 40-60 sr for a Saharan aerosol plume and mixed dust with smoke near Cape Verde, and magnitudes around 42 sr for a Patagonian dust plume. Similarly, Burton et al. (2013) document

inter-quartiles lidar ratios for pure dust of 45-51 sr, estimated from multiple airborne HSRL observations over the western Atlantic and continental U.S., in agreement with measurements over the Caribbean during SALTRACE (Groß et al., 2015). Consistent with the previous studies, HSRL observations in Groß et al. (2013) yield a mean of 48 sr for Saharan dust at Cape Verde. These studies report lidar ratio for dust slightly higher than the revised value used in CALIPSO V4 for pure dust (44 sr). However, it is important to mention that previous studies typically report lidar

ratios corresponding to pure dust plumes and representing a specific atmospheric layer. While CALIPSO-SODA retrievals are generally lower than HSRL/Raman lidar observations, we note that for the Atlantic region adjacent to Africa (20˚W,15˚N) and the Mediterranean Sea, CALIPSO-SODA = 40-50 sr for daytime retrievals (Figure 14). CALIPSO-SODA dust lidar ratios over the open ocean decrease to values generally below 40 sr. We hypothesize that low dust lidar ratios in our study are in part the consequence of the presence of both marine and dust aerosols in a well-

mixed boundary layer, which CALIPSO identifies as pure dust, even though our calculation accounts for marine aerosols in the lower portion of the boundary layer (2L assumption). In addition, it is somewhat surprising that relatively low dust lidar ratios are retrieved over the Arabian sea, with values around 35 sr. Relatively low CALIPSO-SODA lidar ratio for the Arabian Sea region is also supported by Müller et al. (2006) who found dust lidar ratio of 38 sr over the Indian Ocean (Maldives) during INDOEX, but it is unknown from suborbital remotely sensed data whether

low lidar ratios over ocean are a climatological feature. It is interesting to note that evidence of an eastward decrease in lidar ratio over Africa is reported by Schuster et al. (2012) from AERONET land observations, with lidar ratios that decrease from 55 sr over western Africa to values of 40-45 sr in the Middle East. Additional support for source-dependent lidar ratios can be found in Nisantzi et al. (2015) for dust events over Ciprus, where lidar ratio ranges



between 43-58 sr for plumes originated from the Sahara, and 33-48 sr for dust advected from Middle East. Lastly,

evidence of low dust lidar ratio for the Middle East are also reported in Filioglou et al. (2020), with values of $39 \pm 10$ sr for a rural site at the United Arab Emirates.

Kim et al. (2020) exploit the synergy between MODIS AOD and CALIPSO to estimate dust lidar ratios and found median magnitudes of 39.5 sr over the ocean, in relative agreement with the findings in our study. Liu et al. (2011) derived lidar ratios from CALIPSO integrated attenuated backscatter for opaque layers and found median dust

lidar ratio of 36.4 sr for the northeast Atlantic, which is likely biased low (~10%) as the calculations did not account for multiple scattering. Overall, the CALIPSO-SODA median lidar ratio is smaller than the median lidar ratio above clouds of 44.4 sr, estimated by constraining CALIPSO attenuated backscatter with above-cloud AOD (Liu et al., 2015), which in turn is primarily a function of the layer-integrated volume depolarization ratio (Hu et al., 2007).

The paucity of Raman and HSRL observations over oceanic regions currently limits our ability to

comprehensively characterize the changes in lidar ratios throughout a full dust plume lifecycle of mobilization, lofting, transport, and eventual sedimentation. CALIPSO-SODA offers one of the few available satellite datasets that can help track the evolution of dust on a global scale. Future studies that combine CALIPSO-SODA, CALIPSO V4 and back-trajectories derived from meteorological reanalysis and chemical transport models will be critical for separating different aerosol plumes based on their origin and temporal evolution. Future work can also be guided with CALIPSO

depolarization ratio to assess lidar ratio variations as aerosol type transitions from pure to mixed dust.

### 4.3. Polluted continental/smoke and elevated smoke

Daytime polluted continental/smoke lidar ratio (Figure 16) features the most disparate values relative to V4, with open ocean lidar ratios < 45 sr, which is 25 sr less than the value of 70 sr used in V4. In contrast, its nighttime

counterpart generally fluctuates between 55-70 sr. It is worth pointing out that CALIPSO and SODA AOD are virtually uncorrelated for polluted continental during both day and night, indicating that what V4 classifies as "Polluted Continental" is really multiple species with multiple lidar ratios. This lack of correlation will occur for any chosen lidar ratio of this species. In terms of elevated smoke, daytime and nighttime exhibit peaks over the southeast Atlantic (75 sr), the main oceanic region dominated by biomass burning aerosols. More specifically, mean and standard deviation

(preceded by ±) lidar ratios for the peak period of the biomass burning activity from July to October over the oceanic box off the west coast of southern Africa (0˚-20˚S, 5˚W-15˚E) are 69.6 sr ± 12.3 sr and 71.2 sr ± 11.5 sr for day and night, respectively. For other oceanic regions, lidar ratio varies between 35-50 sr, with a local minimum over the east coast of North America and the tropical Pacific (45 sr). Low lidar ratios for smoke are typically observed over the open ocean, which is possibly indicative of aerosol misclassification, however, testing this hypothesis warrants more

investigation. When the analysis is limited to relatively thick aerosol layers (AOD > 0.15, Figure A3 and A4), lidar ratios remain consistently greater than 60 sr because samples are generally limited to the southeast Atlantic and coastal regions (Figure A4). Unfortunately, the calculation of lidar ratio statistics and maps using 5-km classification samples for polluted continental and elevated smoke is not possible owing to both their limited occurrence of 5-km samples and the constraint of limiting the analysis to profiles with only one aerosol type.




## 5. Concluding Remarks

We report one of the first comprehensive studies that examines the aerosol lidar ratios assumed by CALIPSO algorithm for the determination of AOD and aerosol extinction, using an independent CALIPSO research product, CALIPSO-SODA. CALIPSO-SODA algorithm solves the lidar equation constrained with a CALIPSO-CloudSat-based
AOD (SODA, Josset et al., 2015), and thus, a priori information about the aerosol type is not required. Because CALIPSO-SODA lidar ratio is representative of the atmospheric column, we assess CALIPSO V4 for retrievals in which the profiles are characterized by only one aerosol type.

- Profiles with aerosol classification derived from 80-km spatially averaged measurements yield CALIPSO-SODA lidar ratios in disagreement with expectation, with type inter-differences within 6 sr and 17 sr for
daytime and nighttime observations, respectively, and thus, substantially differing from the 46 sr range in the prescribed lidar ratios of V4.

- For aerosols classified from 5 and or 20 km averaged V4 data (i.e. 80-km excluded), CALIPSO-SODA lidar ratios are within ±10 sr those assumed by V4, except for polluted continental aerosol and elevated smoke, which feature, respectively, daytime values around 40 sr and 57 sr, whereas V4 prescribes values of 70 ±
25 sr and  70 ± 16 sr. The CALIPSO-SODA lidar ratios associated with horizontal averaging of 20 and/or 5 km are somewhat consistent with the prescribed V4 values for clean marine aerosols over open ocean and the Southern Ocean (20-25 sr). In addition, for the best quality observations and in regions with well-known dominant influence of specific aerosol species, median CALIPSO-SODA lidar ratio is close to the value assumed by V4 within 5 sr. This is the case for dust over the northeast Atlantic (40-50 sr) which compares
favorably with V4 (44 ± 9 sr), although lower values over other regions suggest a mixture of dust and marine aerosols. For elevated smoke over the Southeast Atlantic Ocean, lidar ratios are within 65-75 sr, close to observational expectations (e.g. Burton et al., 2013, Groß et al., 2013), and in good agreement with V4 value (70 ± 16 sr). In addition, samples classified as elevated smoke are generally confined to coastal areas with AOD > 0.15. In regions where the transport of continental pollution over the ocean is a climatological
feature (e.g. coastal China and the Bay of Bengal), clean and dusty marine types are most likely a mixture of marine aerosol and pollution.

- Discrepancies between CALIPSO-SODA and V4 lidar ratios are attributed to two primary sources: misidentification of aerosol types, and incorrect characterization of the range of lidar ratios (i.e., S ± ΔS) spanned by specific aerosol types. Namely, we attribute substantial differences between estimated lidar ratio
and the prescribed value in V4 to issues with aerosol classification. This appears to be the case for daytime polluted continental/smoke, characterized by CALIPSO-SODA lidar ratios that are 30 sr smaller than the one used in V4.  The misidentification of daytime polluted continental aerosol could be possibly reduced by modifying the depolarization ratio threshold that separates this type from dusty marine in Table 1 (which would result in an increase of dusty marine samples). Moreover, a more comprehensive analysis of
depolarization ratio would be beneficial for refining the definition of dusty marine, as similarities in





magnitude and spatial distribution of lidar ratio between clean and dusty marine aerosols suggests that some dusty marine samples (according to V4) should be classified as clean marine.

- Daytime and nighttime differences are consistent with larger uncertainties in daytime retrievals contributed by the lower daytime SNR associated with the solar background component. Its most dramatic effect is observed for daytime polluted continental/smoke, characterized by a median CALIPSO-SODA at least 25 sr less than the V4 assigned. In contrast, its CALIPSO-SODA nighttime counterpart is 15 sr less than V4, and in better agreement for samples with AOD > 0.15. It is however during nighttime, when smoke lidar ratio compares less favorable with the expected value of 70 sr, and associated with the contribution of areas away from the continents which were not sampled during daytime, likely due to their low AOD and low SNR.



- An aspect not addressed in the current version of CALIPSO-SODA is how to characterize in the algorithm the lidar ratio for lower-tropospheric layers not detected by CALIPSO. Although this is partially resolved by the 2L assumption for the atmospheric mixed layer, profiles in which the lowest aerosol base height derived by CALIPSO is well above 2 km (e.g. dust and elevated smoke) need a rather different approach. Similar to 2L, an alternative solution would be to use a constant lidar ratio for the altitudes below the lowest aerosol base height, which can be determined from CALIPSO-SODA. In this regard, profiles in which the uppermost layer top height is below 2 km yield a median CALIPSO-SODA lidar ratio of approximately 30 sr irrespective of the aerosol type. A modification of the 2L method by including an additional lidar ratio above the mixed layer would produce an overall increase of lidar ratio. This change is expected to be modest and confined to a few sr as the AOD allocated to the undetected layers is likely a small fraction of the total. Yet, these small variations will likely yield better consistency between CALIPSO-SODA and V4 lidar ratios. Lastly, other promising satellite methods that use AOD retrievals over liquid clouds for deriving lidar ratio (e.g. Liu et al., 2015) are advantageous as they circumvent the problem of prescribing the lidar ratio in the boundary layer, especially for elevated aerosol plumes. However, uncertainties in retrieved lidar ratios require a better characterization of potential errors in above-cloud AOD retrievals, through inter-comparison of different satellite products, aided with airborne lidar observations particularly over the eastern Atlantic (e.g. Redemann et al., 2021).




**Data availability.** CALIPSO version 4.2 is available at https://eosweb.larc.nasa.gov (last access: August 24, 2020)
– https://doi.org/10.5067/CALIOP/CALIPSO/LID_L2_05kmAPro-Standard-V4-10 (Vaughan et al., 2019b), and SODA aerosol optical depth is available at http://www.icare.univ-lille1.fr/projects/soda (last access: July 13, 2020; Josset et al., 2015). CALIPSO-SODA lidar ratio is available at:  sftp **login**@xfr999.larc.nasa.gov

**Author contributions.** DP and GS designed the study, ZL conducted the analysis, and MC produced the CALIPSO-SODA retrievals. ZL and DP wrote the paper with contributions from all the co-authors.

**Competing interests.** The authors declare that they have no conflict of interest.

**Financial support.** This research has been supported by the CALIPSO program.





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

***Tables***

*Table 1 classification criteria for tropospheric aerosols in CALIOP V4 over the ocean.*

| | Depolarization ratio at 532 nm | Integrated Attenuated backscatter at 532 nm | $Z_{top}$ (km) | $Z_{base}$ (km) | V4 $S_a$ (sr) |
|---|---|---|---|---|---|
| Clean marine | ≤0.075 | >0.01 | ≤ 2.5 | | 23±5 |
| | <0.05 | ≤0.01 | ≤ 2.5 | | 23±5 |
| Dust | >0.2 | | | | 44±9 |





| | | | | |
|---|---|---|---|---|
| *Polluted continental/smoke* | ≥0.05 and ≤0.075 | ≤0.01 | ≤2.5 | *70±25* |
| *Polluted dust* | >0.075 and ≤0.20 | | >2.5 | *55±22* |
| *Elevated smoke* | ≤0.075 | | > 2.5 | *70±16* |
| *Dusty marine* | >0.075 and ≤0.20 | | ≤2.5 | *37* |

*Table 2: AOD statistics: V4-SODA absolute and relative biases (in %) root mean square difference (RMSE) and fractional RMSE (in %), and linear correlation coefficient r.  Percentage values are relative to the mean SODA AOD.*

| | Day | | | | | Night | | | | |
|---|---|---|---|---|---|---|---|---|---|---|
| | *SODA AOD* | *Bias* | *Bias (%)* | *RMSE* | *RMSE (%)* | *r* | *SODA AOD* | *Bias* | *Bias (%)* | *RMSE* | *RMSE (%)* | *r* |
| *Clean Marine* | 0.10 | -0.02 | -20 | 0.10 | 106 | 0.23 | 0.09 | -0.01 | -13 | 0.10 | 113 | 0.19 |
| *Dusty Marine* | 0.11 | 0.01 | 9 | 0.18 | 164 | 0.22 | 0.14 | 0.04 | 26 | 0.21 | 149 | 0.28 |
| *Dust* | 0.23 | 0.10 | 45 | 0.38 | 163 | 0.59 | 0.29 | 0.24 | 81 | 0.44 | 149 | 0.63 |
| *Polluted Dust* | 0.18 | -0.10 | -59 | 0.24 | 134 | 0.35 | 0.11 | -0.04 | -33 | 0.24 | 226 | 0.23 |
| *Polluted Cont./smoke* | 0.07 | 0.00 | 6 | 0.13 | 171 | 0.02 | 0.08 | -0.02 | -30 | 0.19 | 240 | -0.01 |
| *Elevated Smoke* | 0.26 | 0.00 | -1 | 0.34 | 131 | 0.31 | 0.22 | 0.28 | 128 | 0.59 | 267 | 0.24 |


*Table 3: Median, mean, and standard deviation of CALIPSO-SODA lidar ratio associated with each tropospheric aerosol subtype. Statistics exclude profiles with 80 km horizontal averaging.*

| **Aerosol type** | **V4** | **Lidar ratio statistics (sr)** | | | | | |
|---|---|---|---|---|---|---|---|
| | | **Day** | | | **Night** | | |
| | | **Median** | **Mean** | **Std** | **Median** | **Mean** | **Std** |
| **Clean Marine** | *23±5* | 30 | 33 | 15 | 29 | 33 | 16 |
| **Dusty Marine** | *37±15* | 33 | 36 | 16 | 32 | 35 | 16 |
| **Dust** | *44±9* | 39 | 42 | 19 | 35 | 37 | 13 |
| **Polluted Cont./smoke** | *70±25* | 43 | 45 | 17 | 56 | 57 | 18 |
| **Polluted Dust** | *55±22* | 54 | 52 | 19 | 50 | 51 | 18 |



| Elevated Smoke | 70±16 | 57 | 55 | 20 | 47 | 47 | 20 |
|---|---|---|---|---|---|---|---|



Table 4: CALIPSO-SODA lidar ratio as in Table 3 but for samples classified by CALIPSO at 5-km spatial resolution.


| Aerosol type | V4 | Lidar ratio statistics (sr) | | | | | |
|---|---|---|---|---|---|---|---|
| | | Day | | | Night | | |
| | | Median | Mean | Std | Median | Mean | Std |
| Clean Marine | 23±5 | 24 | 26 | 11 | 25 | 28 | 12 |
| Dust | 44±9 | 34 | 39 | 22 | 34 | 35 | 10 |
| Dusty Marine | 37±15 | 24 | 27 | 13 | 28 | 30 | 12 |









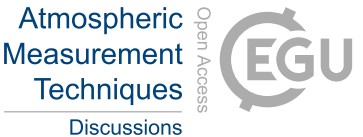

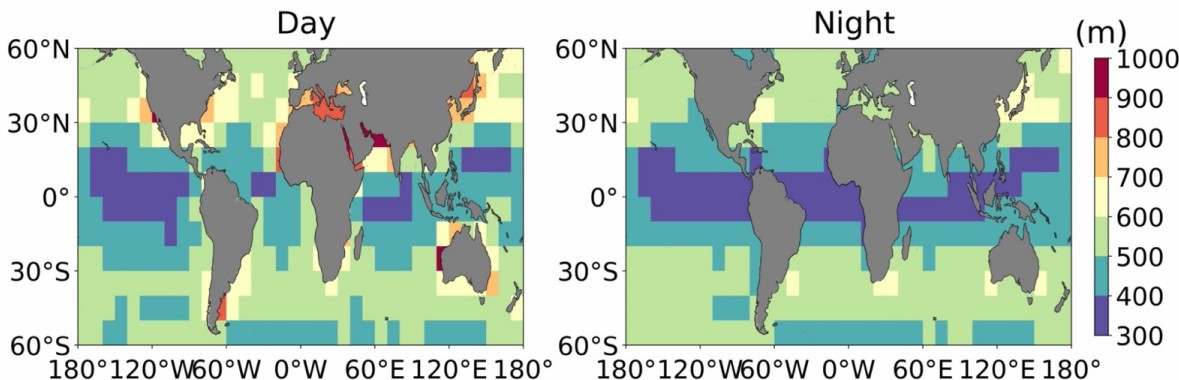

*Figure 1: Marine atmospheric boundary layer height for the period of study estimated from GEOS-5 for daytime (left) and nighttime (right).*


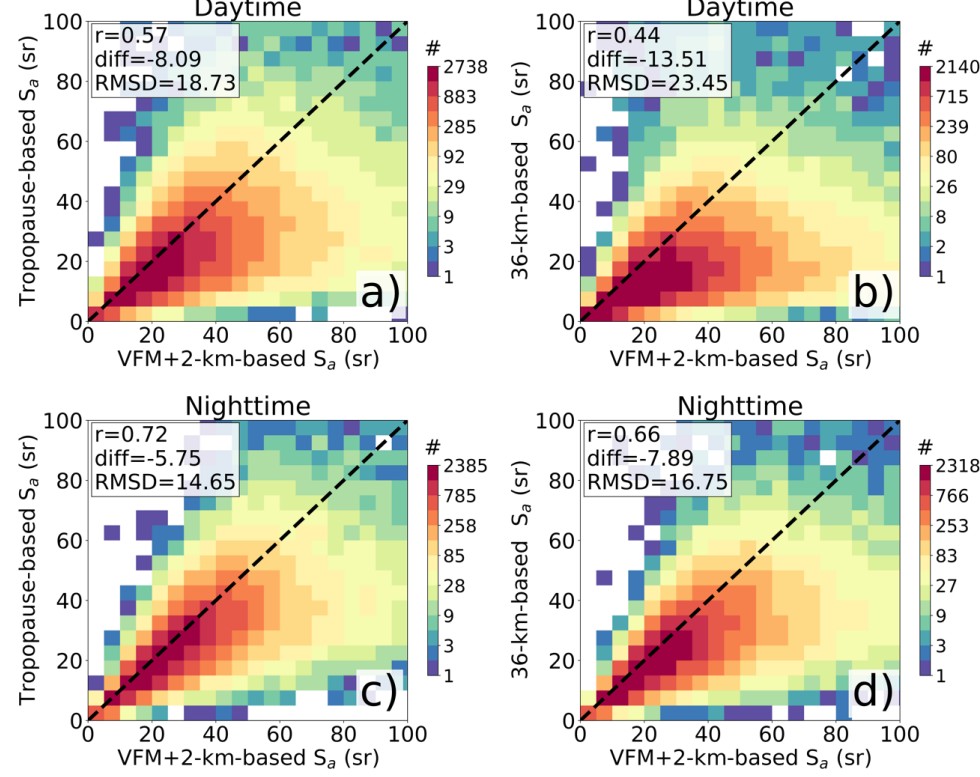

*Figure 2: Inter-comparison between CALIPSO-SODA lidar ratio derived from the standard assumption for initial altitude (VFM+2km, VFM-based) and those estimated using the tropopause height (a and c) and 36-km altitude (b and d). Figures are constructed from 5 days of CALIPSO overpasses during July 2010.*






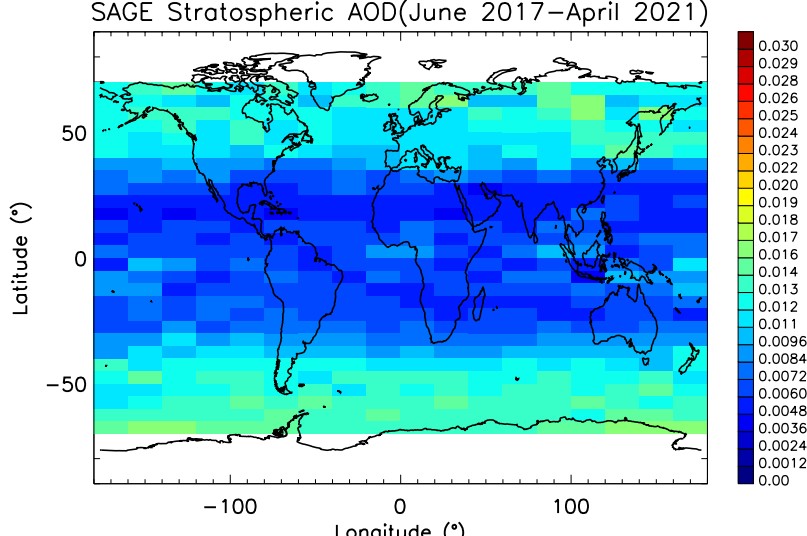


Figure 3: Stratospheric AOD (532 nm) climatology from SAGE-III for the period June 2017-April 2021.





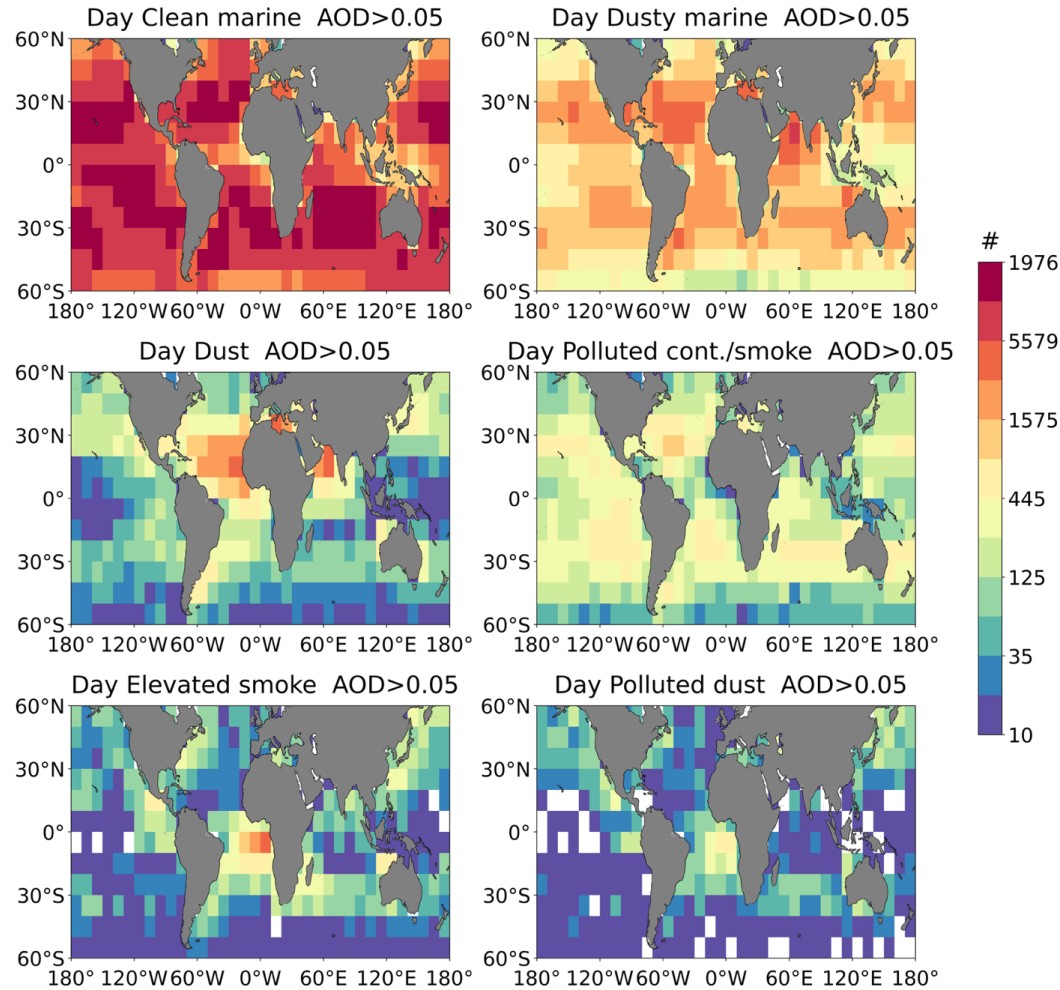


*Figure 4: Geographical distribution of daytime number of samples used in this study.*



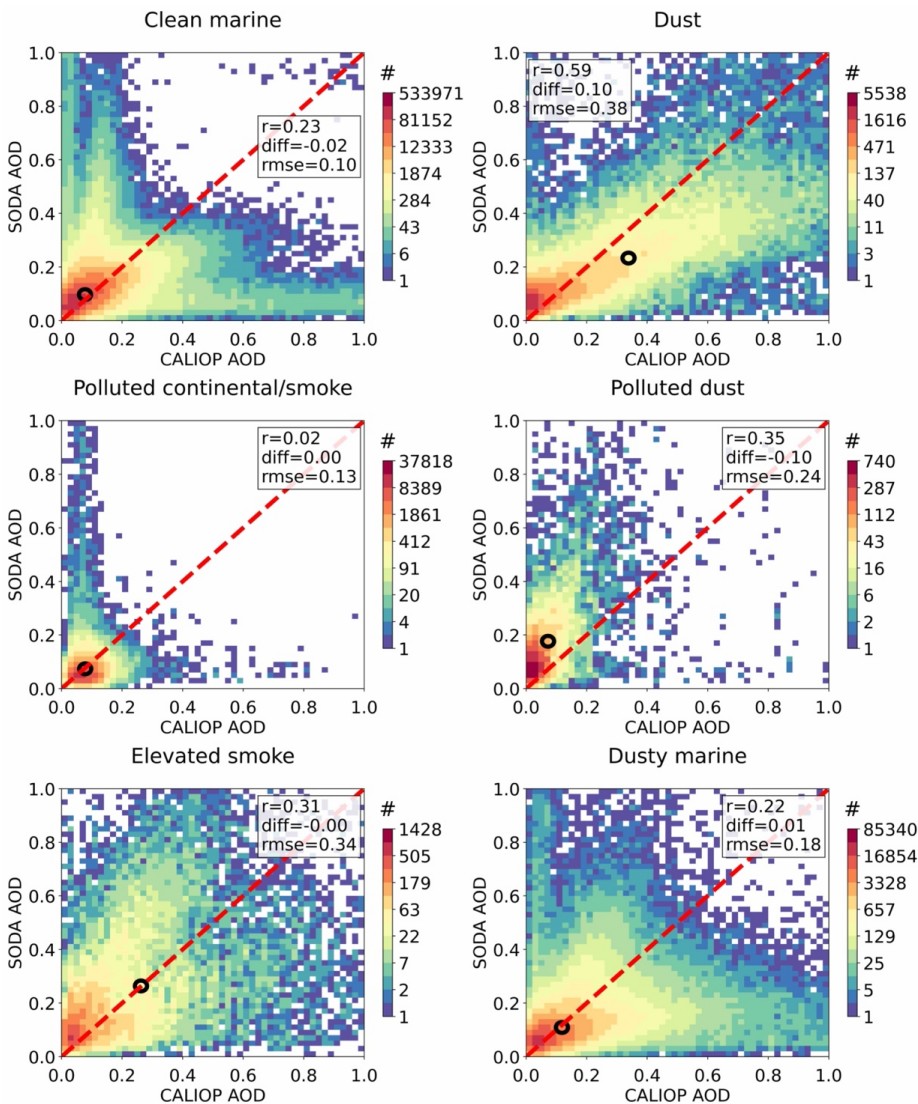

*Figure 5: Bivariate histograms between daytime SODA and CALIPSO AOD for the six aerosols species over the ocean. Black circles represent the mean SODA and CALIPSO AOD.*




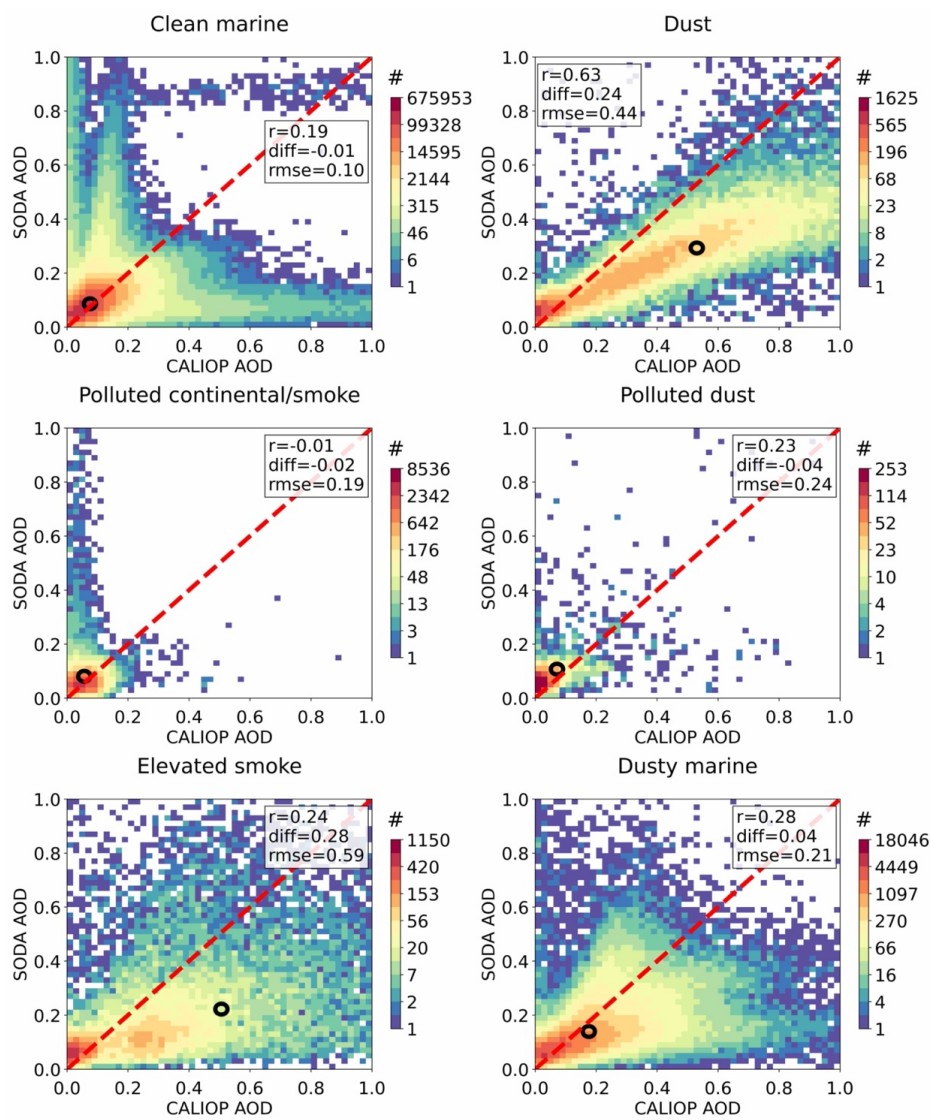

Figure 6: Bivariate histograms between SODA and CALIPSO AOD but for nighttime retrievals. Black circles represent the mean SODA and CALIPSO AOD.








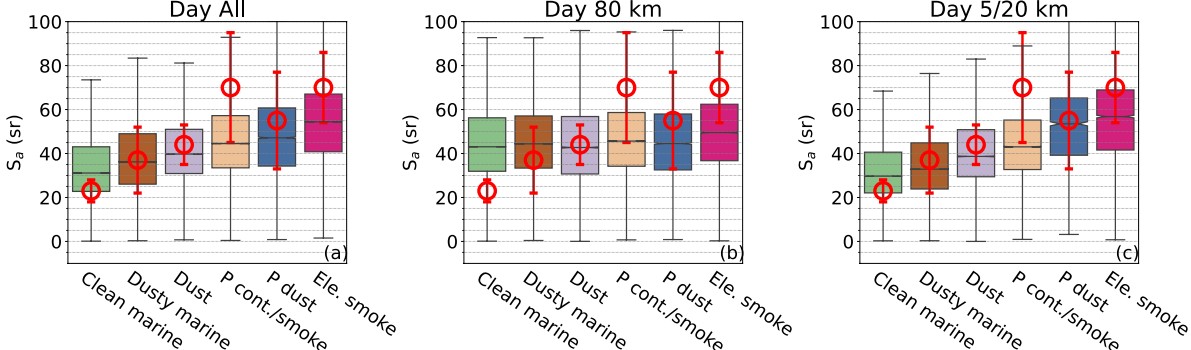

*Figure 7: CALIPSO-SODA Lidar ratio for each aerosol type depicted as a notched whisker box plot. Median values are represented by the horizontal line within each box (also provided in Table 3), and the edges indicate the lower and upper quartile (25% and 75%). Notches (sometimes negligible) represent the 95% confidence interval of the median. Error bars denote upper and lower 0.7 % of a Gaussian distribution. Red circles represent the prescribed V4 lidar ratios for each aerosol type (the six values repeat in each plot) with error bars denoting the associated uncertainty. a) irrespective of the CALIPSO horizontal average, b) profiles that contain only CALIPSO 80-km spatial averaging, and c) profiles with 5 and/or 20-km spatial averaging.*

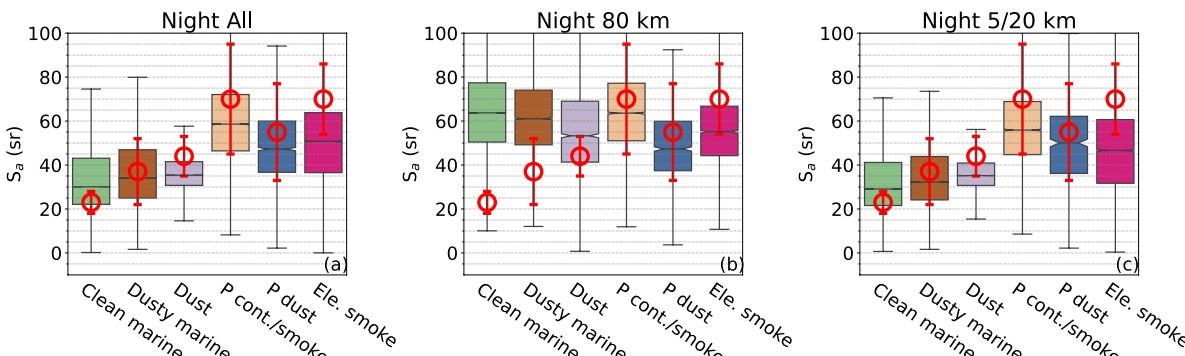

*Figure 8: Lidar ratio as in figure 7 but for nighttime. a) irrespective of the CALIPSO horizontal average, b) profiles that contain only CALIPSO 80-km spatial averaging, and c) profiles with 5 and/or 20-km spatial averaging. Legends and description of the whisker box plot are described in Figure 7.*




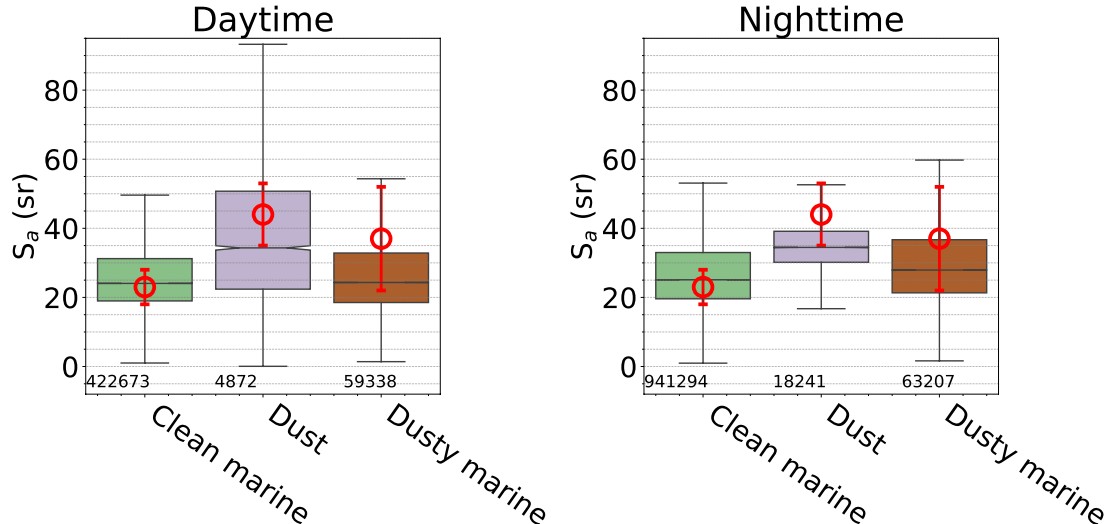

Figure 9: CALIPSO-SODA lidar ratios at 5km horizontal resolution for the three most abundant aerosol types: clean marine, dust, and dusty marine. Symbols and legends as in Figures 7 and 8.






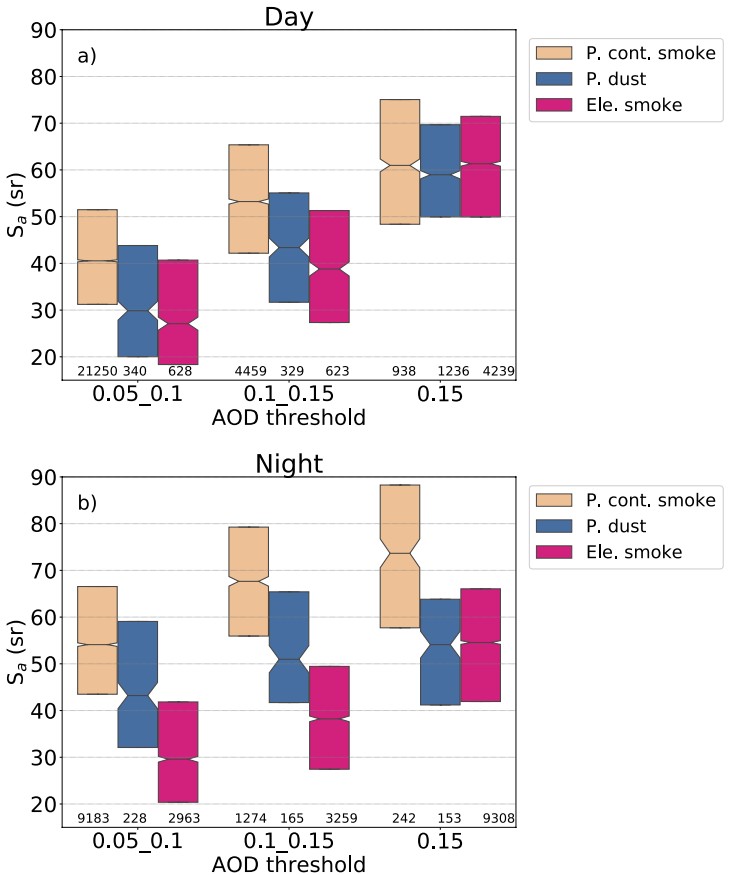

*Figure 10: Median lidar ratios for AOD thresholds of 0.05,0.1, and 0.15 for daytime (above) and nighttime (below) retrievals. Median lidar ratio changes with AOD for dust and marine types remain within 6 sr (not shown).*


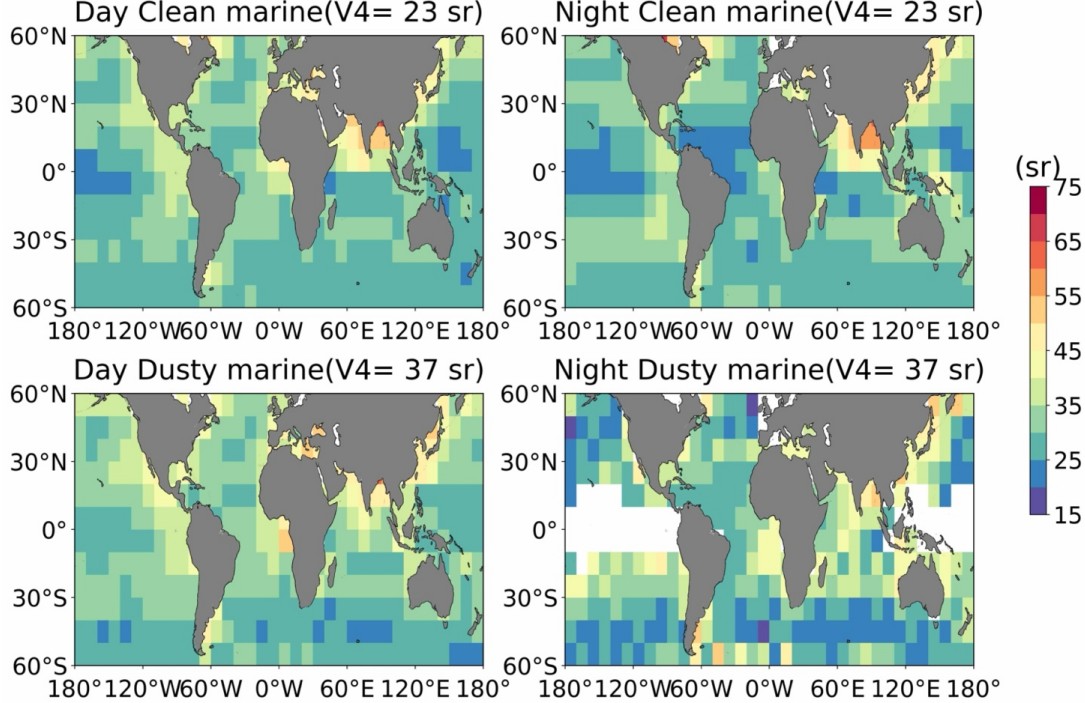

*Figure 11: Median CALIPSO-SODA lidar ratio for clean marine (upper maps) and dusty marine classified from 5-km and/or 20-km spatially averaged data. Maps are constructed with 10°x 10°regular grids, daytime and nighttime*
*(left and right panels, respectively). Values are reported for grids constructed with at least 20 samples.*





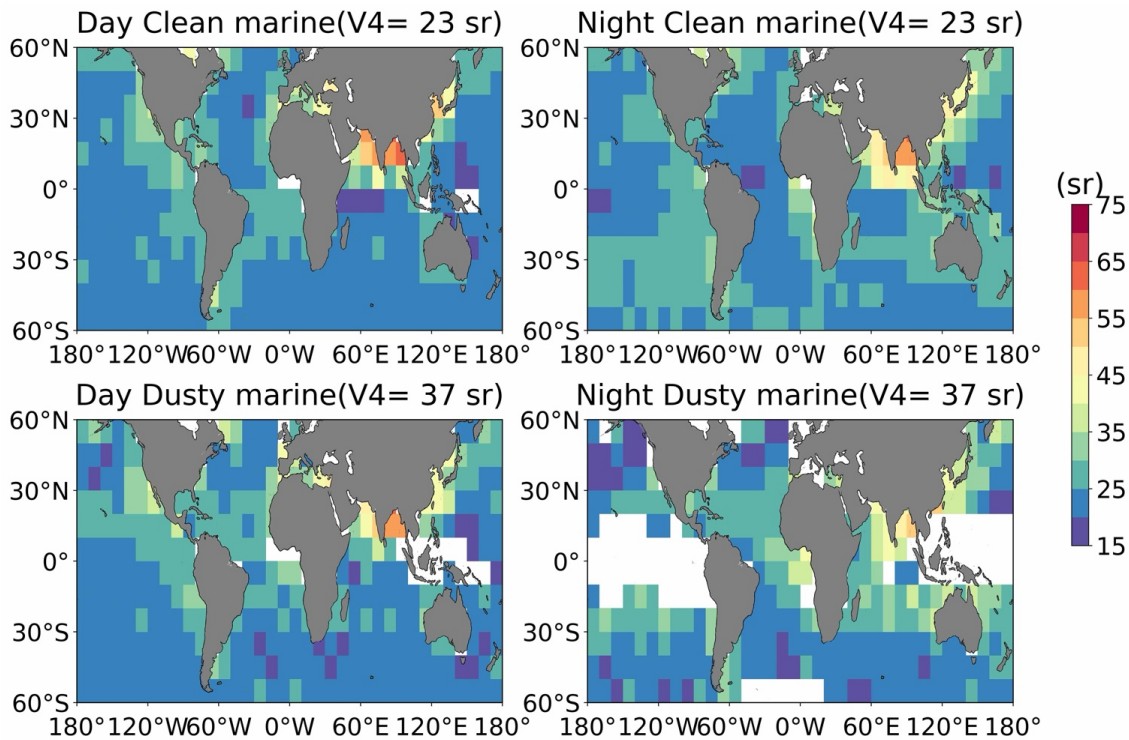

*Figure 12: Median CALIPSO-SODA lidar ratio as in Fig. 8 but for clean marine (upper panels) and dusty marine (lower panels) classified from 5-km spatially averaged data and gridded at 10˚x10˚ resolution.*

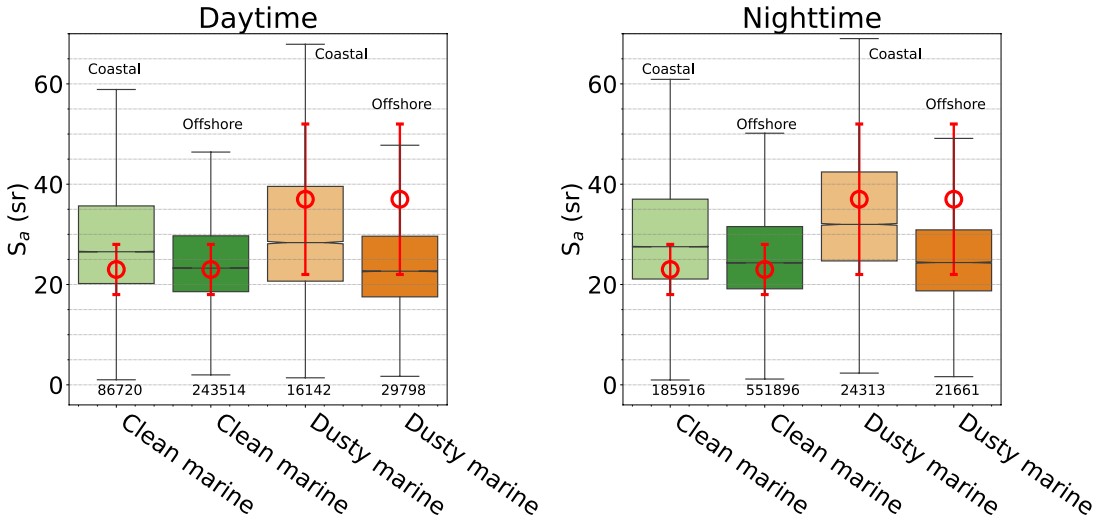

*Figure 13: Lidar ratios for clean marine and dusty marine over oceanic regions within 5˚ degree from the coast (coastal) and at least 10˚ away from the coast (offshore). Red circles represent the prescribed lidar ratio in V4 and the associated uncertainty (error bar). Left and right panels correspond to daytime and nighttime retrievals, respectively.*






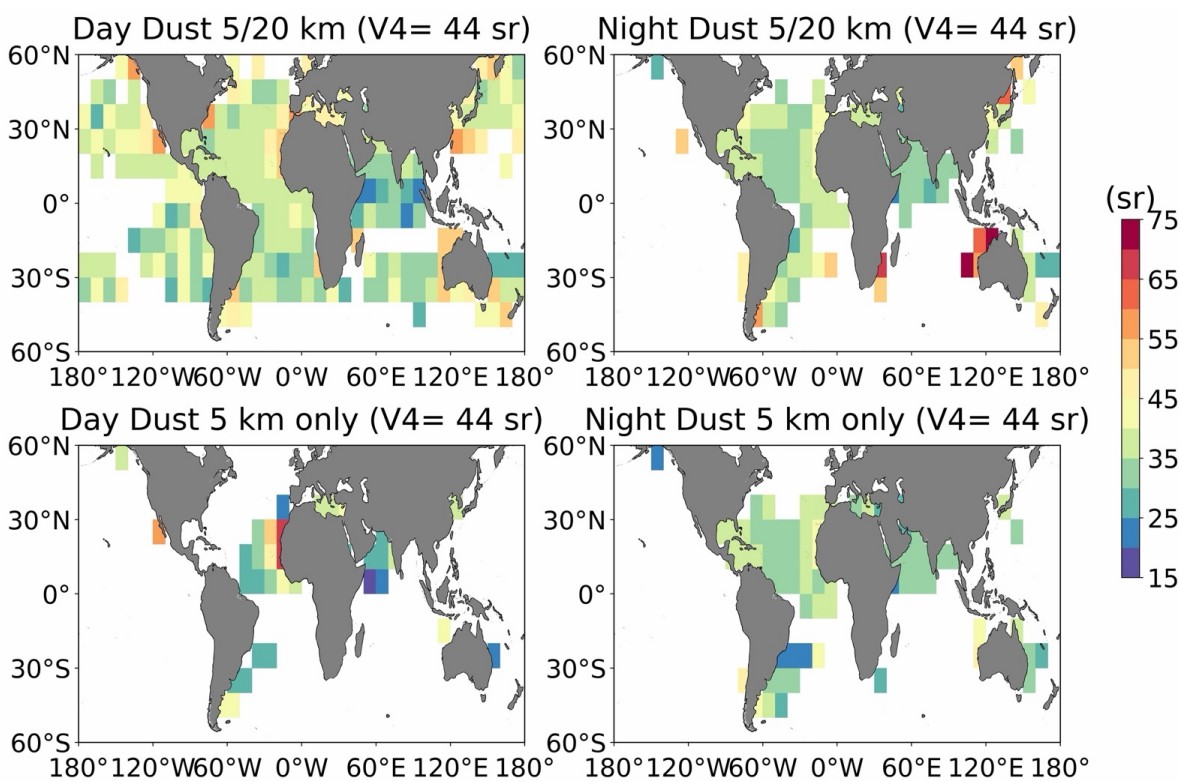

Figure 14: Daytime and nighttime CALIPSO-SODA lidar ratio maps for dust constructed using aerosol typing classification at 5km and/or 20 km resolution (upper panels), and 5 km only (lower panels). Values are reported for grids constructed with at least 20 samples.

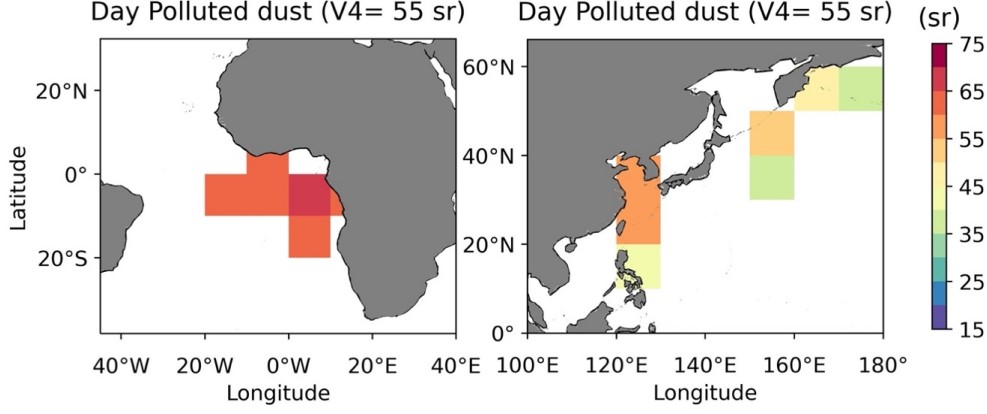

Figure 15: Daytime CALIPSO-SODA daytime lidar ratio for polluted dust classified from 5- km and/or 20-km spatially averaged data for the two regions with available observations: eastern Atlantic (left) and northwest Pacific (right). Values are reported for grids constructed with at least 20 samples.




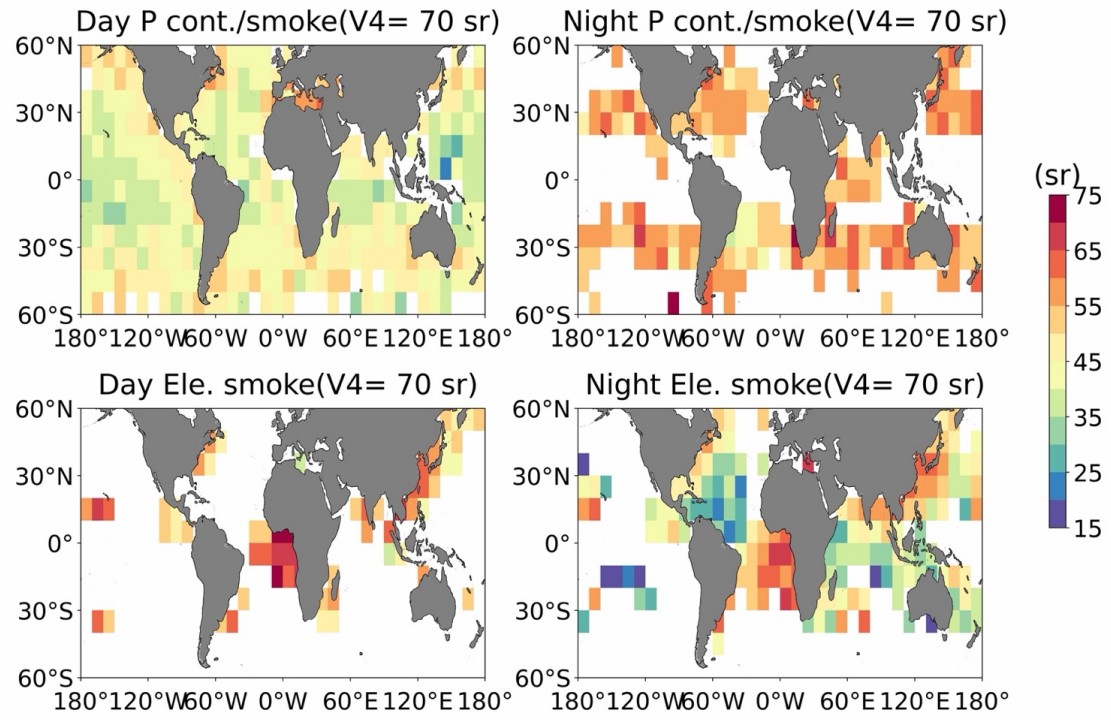


*Figure 16: Lidar ratios as in Fig. 11 but for polluted continental/smoke (upper) and elevated smoke (lower), classified from 5- km and/or 20-km spatially averaged data. Values are reported for grids constructed with at least 20 samples.*









# Appendix

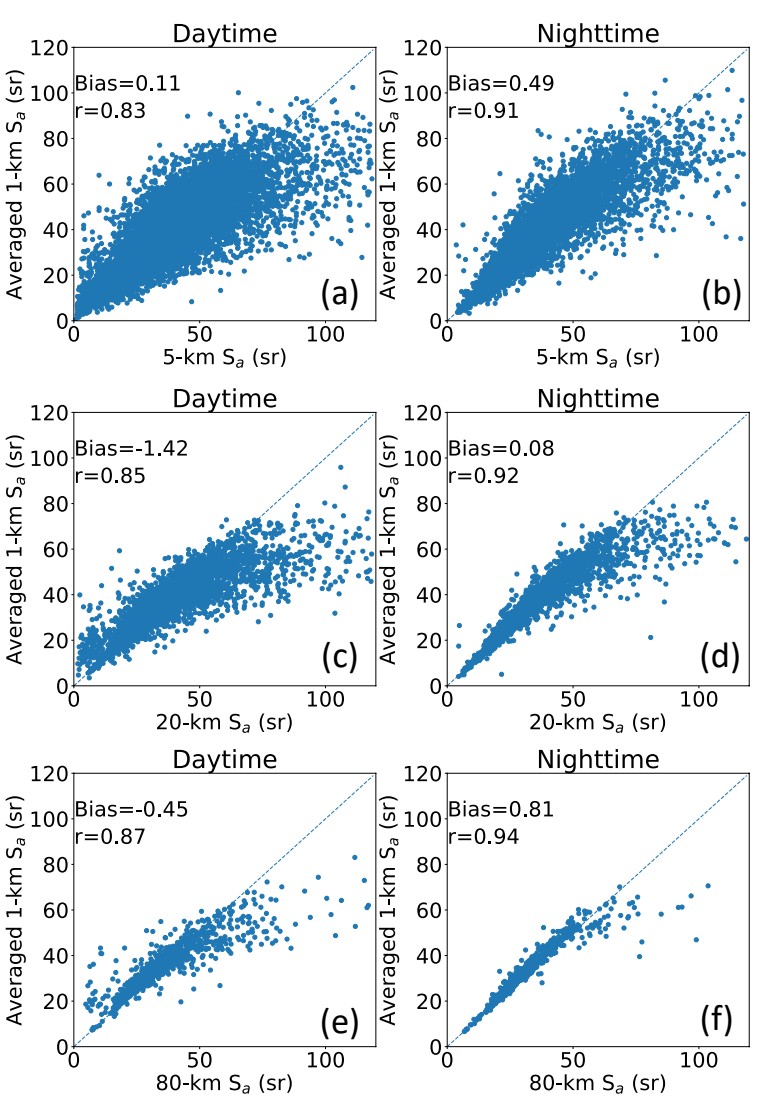


*Figure A1: Average-then-retrieved versus retrieved-then-average methods for lidar ratio. Intercomparison
between lidar ratio retrieved using attenuated backscatter horizontally averaged to 5-km, 20-km, and 80-
km along-track grid size, and lidar ratio derived using 1-km averaged attenuated backscatter, and further
averaged to match the 5-km, 20-km, and 80-km resolution. Data points are showed for samples for which
at least 50% of 1-km retrievals were successfully computed for a given spatial scale.*







*Figure A2: Geographical distribution of nighttime number of samples used in this study.*







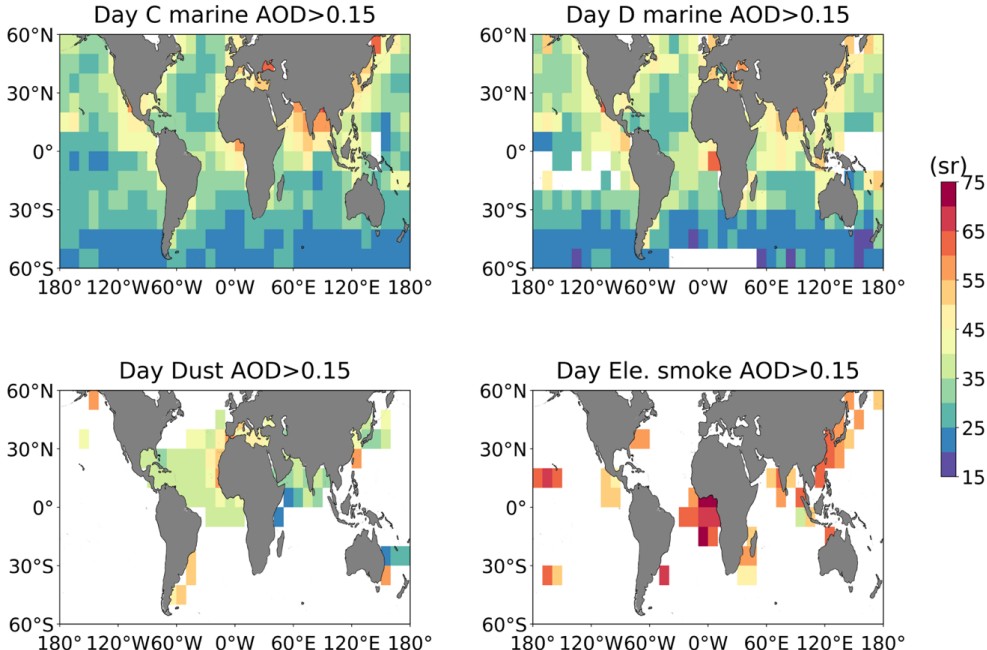

*Figure A3: Daytime median CALIOP-SODA lidar ratios of data associated with AOD greater than 0.15 within 10°x 10°grids for four tropospheric aerosol types that contain sufficient number of samples.*







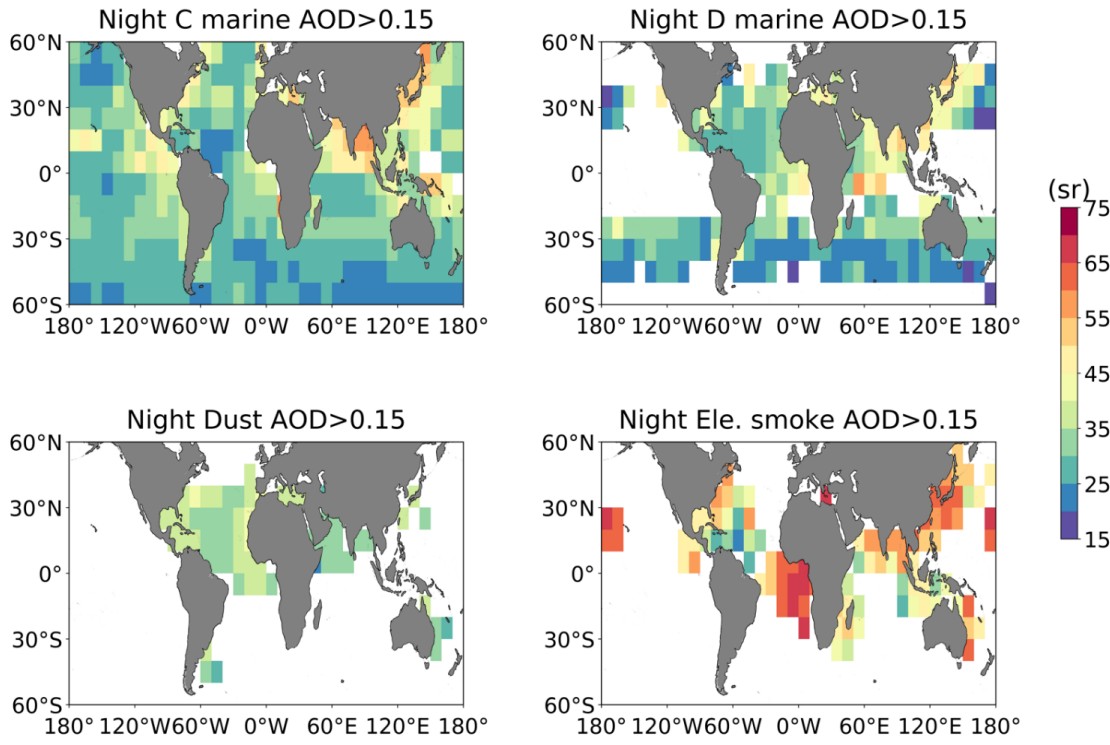

*Figure A4: Same as Figure A3 but for nighttime.*
