# Peer review of "Assessment of tropospheric CALIPSO Version 4.2 aerosol types over the ocean using independent CALIPSO-SODA lidar ratios"

_Atmospheric Measurement Techniques, 2021_

## Author Comment (AC1)

**Dr John Reagan**

This is a most comprehwensive, well written paper that carefully assesses the data and enumerates limitations of various assumptions made in the data assessment.  It presents the best assessment/prediction to date of retrieved aerosol lidar ratios and AODs obtained from CALIPSO observations, made possible using the independent CALIPSO-SODA lidar ratio retrievals for comparison.  Unceertainties in the CALIPSO Version 4.2 and the CALIPSO-SODA retrievals, and biases they contribute, are carefully considered.  Discussions about agreements/disagreements in the comparison results are well reasoned and extensive.

R: We appreciate Dr Reagan's kind words. His recognition of our work is greatly appreciated.

---

## Author Comment (AC2)

**Referee 2**

The submitted paper titled "Assessment of tropospheric CALIPSO Version 4.2 aerosol types over the ocean using independent CALIPSO-SODA lidar ratios" retrieved aerosol extinctions and lidar ratios from a synergetic use of CALIOP L1 profiles and CALIPSO-SODA AODs then assessed lidar ratios used in the CALIOP version 4 aerosol algorithm. The results of this study are very meaningful and may be very helpful to develop further version of the CALIOP aerosol retrieval algorithm. The manuscript is well organized but needs minor revision before publication.

R: We appreciate the reviewer's positive comments and suggestions. Our responses are highlighted in blue.

1) line 55-56: better to move to Section 2 (method). If you declare that you used data for only over ocean, you don't need to mention about plluted dust over desert land in line 65.

Following the reviewer's suggestion, we have moved line 55-56 to section 2 and removed the explanation about polluted dust classification over desert.

2) line 81: So, this is your motivation of this study which is very simple. please describe limitations of previous studies in more detail to emphasize this study. You mentioned the spatiotemporal coverage is extremely limited. Is there any other related studies covering larger regions (global) or long period of time? Your study show any spatiotemporal vatiations which are not discussed in previous studies? Considering a long introduction, your motivation is too simple.

This is, to the best of our knowledge, the first time in which independent lidar ratios are used to characterize the six aerosol types over the ocean. Moreover, the reported lidar ratios (and spatial distributions) in our paper are a unique contribution of Li et al. (2021). Three previous studies have only focused on either specific regions or an individual aerosol types, with a different dataset as the one reported here. We have added the following paragraph to emphasize the knowledge gaps that motivate our much-needed studies:

"…While a few satellite-based studies have endeavored to quantify lidar ratios, they have focused on specific aerosol types, namely, dust (e.g. Liu et al., 2015; Kim et al., 2020) and clean marine aerosols (Dawson et al., 2015). A global assessment of lidar ratio for the six CALIPSO aerosol types over the ocean is, thus, lacking. A detailed lidar ratio characterization is central for refining lidar ratio lookup tables in future CALIPSO versions, as well as potentially improving the aerosol classification scheme, with the final goal of producing more accurate retrievals of aerosol extinction coefficient and optical depth.

In this study, we compare CALIPSO version 4.2 aerosol products and lidar ratios to a CALIPSO-based research product: the CALIOP Synergized Optical Depth of Aerosols (CALIPSO-SODA). We derive the CALIPSO-SODA lidar ratios by applying a Fernald-Klett inversion (Fernald, 1972; Fernald et al., 1984; Klett, 1985) to the CALIOP attenuated backscatter coefficients and the SODA AODs. Our goal is to determine how well the prescribed CALIPSO V4 lidar ratios compare to the retrieved CALIPSO-SODA

lidar ratios for each CALIPSO aerosol type over the ocean. In addition, we are interested in analyzing the spatial variability in lidar ratio for each aerosol type and providing global maps that can guide future improvements of lidar ratio selection for CALIPSO products. "

3) line 89: You have introduced only level 2 product. What about the CALIOP level 1 product used in your study? You had used VFM for cloud screen only for the range of 1

The reviewer is correct. We have added the following information. "Lidar attenuated backscatter is taken from CALIPSO Level 1B, as described in Painemal et al. (2019)".

4) - 8.2 km? what about clouds out of that range? The title of subsection is "CALIPSO V4 and SODA data". But there is nothing about SODA.

An effective range up to 36 km of CALIPSO makes possible detecting clouds at virtually any altitude. We have slightly modified the sentence to read: "cloud mask with 333 m horizontal resolution below 8.2 km and 1 km above, up to 36 km.
SODA product is explained in section 2.3. In the revised section, we have moved the SODA description to section 2.1. (CALIPSO V4 and SODA data).

5) line 117: You mean that any aerosol layers below BL are considered as clean marine, even if the CALIOP algorithm classify as other aerosols such as dusty marine, dust, or polluted continental? It could be acceptable for remote ocean but may wrong for coastal regions. How can you justify this?

Computation of CALIPSO-SODA lidar ratios are independent of aerosol typing, which is why L2 makes the general assumption that the BL lidar ratio is clean marine. In other words L1 and L2 are designed to work independent of CALIPSO aerosol retrievals, especially when no aerosol classification is provided by V4. However, we apply both L1 and L2 assumptions to our analysis, depending on whether a given aerosol type is likely observed in the boundary layer or the free troposphere (Table1). This is why we wrote in the original submission: "…Finally, given that CALIPSO aerosol typing depends on the aerosol layer height (Table 1), we characterize clean marine, dusty marine, and polluted continental smoke using CALIPSO-SODA lidar ratios based on the 1L assumption; dust, polluted dust, and elevated smoke aerosols are described by means of the CALIPSO-SODA 2L assumption, to isolate the lidar ratios from elevated layers from those in the boundary layer (likely dominated by marine aerosols)." While dust is a CALIPSO aerosol type that can occur both in the boundary layer and the free troposphere, the lowest aerosol layer height is typically above the computed boundary layer height (line 123 in the original submission). We have noticed that in some specific areas, the aerosol base height could be at times within the boundary layer. However, accounting for those cases is challenging because dust in the boundary layer (as seen by CALIPSO) does not necessarily reach the surface. In other words, the 2L assumption offers a tractable way to derive lidar ratio for dust and smoke, which, on average, fits the assumption that aerosol layers overlie the marine atmospheric boundary layer.

To avoid any misunderstanding, we have rephrased the following sentence: "Thus, the 2L technique is applied irrespective of the occurrence of the Clean Marine type in V4." to "Thus, both 1L and 2L techniques are used to compute lidar ratios independent of the V4 aerosol typing. However, as discussed in Section 2.3, we select one assumption over the other depending on the likeliness that a given aerosol type occurs in the boundary layer or free troposphere."

6) line 130-175: Why did you select VFM_max+2km as a top height for the retrieval? Is it only because Painemal et al. (2019) showed best agreement with HSRL for that criteria? What did they compare here in Painemal et al. (2019)? Coulmn AOD from CALIPSO-SODA and HSRL AODs from surface to VFM_max+2km? It should be very careful here because you retrieve only below VFM_max+2km, but CALIPSO-SODA is a coulmn integrated AOD from surface to TOA. You ignored aerosols above the VFM_max+2km (line 157). It could be negligible but you showed the discrepancy up to ~70% (line 154) which is quite large. Based on this results, it looks better to select higher altitude as an upper limit for the retrieval. Authors should expain more acceptable reasons for selecting VFM_max+2km as an upper limit for the retrieval and specify resulting unceartainty in the retrived lidar ratio. Authours mentioned uncertainty of the retrived lidar ratios due to Stratospheric AOD (line 173), but tropospheric aerosols above VFM_max+2km is much more important to discuss.

The rationale was to minimize the effect of profile segments with low signal-to-noise-ratio, which otherwise would had added uncertainties to the retrieved lidar ratios. The VFM_max+2km was indeed adopted because it features the best agreement with the HSRL lidar ratio in Painemal et al. (2019). In the original manuscript, we reported sensitivity calculations by modifying the altitude criterion to VFM_max+1km and VFM_max+3 km, and the differences relative to VFM_max+2km were negligible. Further justification for VFM_max+2km can be found in Kacenelenbogen et al. (2011) who found that CALIPSO (V3) aerosol-layer top height is less than 2km lower than the maximum detectable aerosol retrieval from the HSRL. While the reviewer raises a good point, our interpretation for the lidar ratio difference between VFM_max+2km and that using the entire 36-km column is that the discrepancy is explained by the low SNR at high altitudes of the attenuated backscatter. The fact that the differences is reduced during nighttime is also consistent with higher SNR at night, in the absence of background solar radiation (Figure 2). As we only use tropospheric features for truncating the attenuated backscatter profile, we wanted to have an initial estimate of the effect of stratospheric AOD to bound uncertainties in the retrievals. We agree with the reviewer in that assessing the effect of tropospheric aerosols above VFM_max+2km and the stratosphere is relevant. As global observations of free tropospheric aerosols are unavailable, we can only discuss studies that intercompare episodic lidar observations and CALIPSO retrievals, which show that the contribution of atmospheric aerosols 2km above the VFM_max is generally modest (Burton et al., 2013).

7) Line 209: 2L assumption for polluted dust and elevated smoke is acceptable. however, dust aerosols may frequently exist near the sea surface, especially near the continents. Have you checked this?

See our response to comment 5)

line 236: better not to conclude like this. present uncertainties of each data here.

In the preceding sentences, we briefly summarized intercomparisons between HSRL, MODIS, CALIPSO, and SODA AOD. Based on the close agreement between HSRL, MODIS, and SODA AOD, line 236 is well justified. In the revised manuscript, we have included the mean differences between different products to better justify our statement:

"Similarly, Painemal et al., (2019) found a better match between airborne HSRL and SODA AOD (slope of 0.96) than that for V4 (slopes of 0.71), as well as a better regional agreement with MODIS AOD Collection 6 (mean differences <0.06 and < 0.18 for SODA and V4, respectively)"

line 258-260: "misclassification of ~ tenuous aerosol layers that are not detected by the CALIPSO algorithm" This may be a reason why the authors should select higher upper boundary for the retrieval instead of VFM_max+2km.

The under-detection of tenuous layers is not limited to layers above VFM_max as they can occur at any altitude of the profile. A fundamental problem with including attenuated backscatter at high altitude is the low SNR, which can yield systematic biases in the retrievals. For instance Young et al. (2013) found a positive bias in AOD for samples with low SNR. In other words, extending the profile higher in the free troposphere does not guarantee retrieval improvements. In the revised manuscript, we have added information about the SNR-driven bias discussed in Young et al. (2013).

---

## Author Comment (AC3)

**Referee 3**

**Major comments**

We recommend that the authors:

. first compare CALIOP-SODA and CALIOP V4 AOD as well as lidar ratios for all CALIOP V4 aerosol types before they classify their analysis by CALIOP V4 aerosol types. As they know, CALIOP V4 is likely to miss tenuous aerosols or misclassify aerosols. Starting by classifying CALIOP-SODA lidar ratios per (likely misclassified) CALIOP V4 aerosol type is confusing and slightly circular.

R: We appreciate the reviewer's comments and suggestions for improving our analysis. Our responses are highlighted in blue. While comment 1) is indeed relevant, it is less applicable to our work, as explained in the following. We agree with the reviewer in that aerosol misclassification and under-detection are issues to be taken into account in any CALIPSO-centric analysis. However, the reviewer would agree that addressing these issues in detail from a global-scale perspective is unfeasible, and that assessments of lidar ratio irrespective of aerosol type does not address the main issue tackled in our study, which is intercomparing CALIPSO-SODA lidar ratios with the V4 lookup table for specific aerosol types. Instead, our approach is more pragmatic: we first start by considering that CALIPSO aerosol typing is providing meaningful information which does not necessarily overlap with other typing definitions obtained from suborbital and ground-based lidar ratios (e.g. using HSRL or Raman lidar) but yield aerosol clusters with measurable characteristics. So, our objective here is characterizing the lidar ratio of CALIPSO aerosol type, and provide maps that can be guide the selection of lidar ratios for a future CALIPSO version. We would like to emphasize that our main objective is not to recommend specific refinements to the aerosol classification to achieve more consistency relative to other studies that make use of more advanced lidar measurements. In sum, we are interested in finding ways to select more adequate lidar ratio that can yield better aerosol extinction coefficient and aerosol optical depth. Regarding the misclassification and under-detection, our methodology intends to minimize these uncertainties by using an AOD threshold and performing the analysis when only one aerosol type is observed in the aerosol column.

. clarify why CALIOP does not consider the possibility of polluted dust within the PBL

Polluted dust is defined for layers with base above 2. 5 km because it was found in previous HSRL studies (e.g., Burton et al., 2013) that polluted dust in Version 3 has characteristic lidar ratios of 35 sr, suggesting that aerosols that were actually mixture of dust and marine aerosols were misclassified by V3 as polluted dust in the boundary layer. This finding motivated the inclusion of a new aerosol type in V4: dusty marine. To identify dusty marine aerosol mixtures, polluted dust was redefined as being confined to the free troposphere (>2.5 km), to reduce the number of samples misclassified as polluted dust in the boundary layer. For more details, we refer the reviewer to Kim et al. (2018).

. clarify their filtering method and technique as a bullet list or a table (includes 1L vs 2L techniques, cloud masking, altitude selection etc)

We appreciate this suggestion. In the revised manuscript, we have added the following table to summarize the methodology

*Table 2: Summary of methodology applied to CALIPSO-SODA and V4 intercomparison.*

| Condition | Application |
|---|---|
| AOD threshold | SODA AOD > 0.05 |
| Aerosol types | Over the ocean, with only one aerosol type throughout the column (excluding clear) |
| CALIPSO-SODA 1L assumption | applied to dust, smoke, and polluted smoke |
| CALIPSO-SODA 2L assumption | applied to clean marine, marine dust, polluted dust |
| CAD score | > \|-50\| |
| Cloud coverage | cloud free over the 5-km horizontal resolution |

. clearly suggest which CALIOP product is accurate, which one is not. And proposes fixes to the algorithm or additional filters to be applied by the users moving forward.

Section 5 already provides a summary with the information required by the reviewer. The results are dependent on multiple factors including spatial resolution and geographical location and, thus, it is challenging to say that product A is more accurate than product B. While we did provide recommendations for refining the classification of dusty marine and polluted continental smoke aerosols, improving the typing scheme requires more dedicated efforts beyond the scope of our work. As our work deals with assessing prescribed lidar ratios and motivates future work for refining lidar ratios in a future CALIPSO version, we do not have specific suggestions for CALIPSO users.

**Detailed comments:**

. line 14 – "This implies that the CALIPSO classification scheme generally categorizes aerosols correctly" is too strong of a statement. Please consider watering it down.

We modified the sentence to read: "This implies that the CALIPSO classification scheme generally categorizes specific aerosols types correctly over regions where they are abundant"

. line 21 – "value"

Corrected, thanks.

. abstract – the authors point out issues with CALIOP V4 polluted continental/ smoke, dust, dusty marine and clean marine aerosol types. They should suggest some future fixes and possible filtering

For the sake of conciseness, we have left the abstract as is. Discussion is provided throughout the manuscript.

. line 105 – why not talk about HSRL instead of saying "airborne lidar observations"?

The sentence was slightly modified to: "airborne HSRL observations"

. Figure 1 – can you briefly specify how reliable is GEOS-5 ML?

McGrath-Spangler and Molod (2014) compared several methods for computing planetary boundary layer depth using GEOS-5. They found that the bulk Richardson number method (which is adopted in our paper) provides the best match with radiosonde-based estimates over land. Von Engeln and Teixeira (2013) noted that the bulk Richardson number method yields heights well below the inversion height in cloud-topped marine boundary layers, suggesting that the estimated MBL is more closely related to the cloud base height. The revised manuscript briefly summarizes this information.

. line 226 – I don't understand the sentence. Please rephrase.

We have rephrased the sentence to read:

"Lastly, polluted dust resembles the spatial distribution of elevated smoke, which reflects the influence of biomass burning emissions (especially in the South Atlantic) and that these are the only two aerosol types-defined for aerosol plume elevations above 2.5 km a.m.s.l. when the depolarization ratio is below 0.2 (Table 1)."

. line 240 or eq. (1) – explain i=1 to N

N denotes the number of samples. This information is included in the revised manuscript.

. line 249 – largest mean AOD

The sentenced was modified accordingly.

. line 221 – consider replacing "modest" by low and then "particularly low" instead of "negligible"

Done

. Figure 5 & 6 – consider saying "density plot" instead of histograms

We appreciate the reviewer's suggestions. We believe that the concept of bivariate histogram is more accurate than density plot.

. Figure 7 – add (a), (b) and (c) on graphs

The labels were already included in the original submission

. These Figures and Tables are redundant, consider consolidating/ simplifying – Fig 9 and Table 4 as well as Fig. 7-8 and Table 3

We have tried several combinations of tables and formats, as well as reducing the number of figures. However, it is quite challenging to create compact tables/figures that clearly show the results that we want to highlight.

. Line 275 to 277 – aerosol variability depends on the environment. Please refer to e.g., Shinozuka and Redemann (2011).

We appreciate drawing our attention to Shinozuka and Redemann. We have modified the sentence to read: "This could be in part caused by mixing of different aerosol plumes at such large horizontal scales especially near emission sources (e.g. Shinozuka and Redemann, 2011)…"

. Line 305 – there is a repeat in the sentence. Please rephrase

The sentence was rephrased to read:

    "Interestingly, polluted continental lidar ratio for AOD > 0.15 reaches values near 72 sr, in good

agreement with the value used in V4 (70 sr)."

. Line 303 to 307 – can we make sure to say that the lidar ratio for a specific aerosol type should be independent of the AOD? The fact that it varies with AOD points to some issues in the CALIOP algorithm and I suggest describing them.

Lidar ratio is an intensive parameter that does not correlate with AOD. Certainly, uncertainties in AOD can yield bias in lidar ratio for the same attenuated backscatter profile. However, variations of lidar ratio with AOD observed in our study are primarily related to the fact that optically thicker aerosol layers can be better identified by the typing algorithm, implying that lidar ratios will better match the prescribed lidar ratios for higher AOD.

. Line 432 – assumed by the CALIPSO

corrected

. first bullet in the conclusion – consider explaining the reasons and implications

We have added the following sentence: "Classification issues for 80-km averaged samples are likely, as spatial averaging are performed to increase the SNR for tenuous aerosol layers, rendering more uncertain retrievals than its 5-km and 20-km counterparts."

. Line 443 – within +-10sr of those

corrected

. Line 460 – is a repeat of the sentence above

The sentence was slightly modified to read: "Namely, we attribute substantial differences between estimated lidar ratio and the prescribed value in V4 for polluted continental/smoke to aerosol classification, as the retrieved lidar ratios are 30 sr smaller than the one used in V4. "

Shinozuka, Y. and Redemann, J.: Horizontal variability of aerosol optical depth observed during the ARCTAS airborne experiment, Atmos. Chem. Phys., 11, 8489–8495, https://doi.org/10.5194/acp-11-8489-2011, 2011.